# Targeting of NAT10 enhances healthspan in a mouse model of human accelerated aging syndrome

Gabriel Balmus [1,2], Delphine Larrieu [1,9], Ana C. Barros[1,2], Casey Collins [2], Monica Abrudan [2], Mukerrem Demir[1], Nicola J. Geisler[1,2], Christopher J. Lelliott [2], Jacqueline K. White[2], Natasha A. Karp[2,3], James Atkinson [4], Andrea Kirton[2], Matt Jacobsen [4], Dean Clift[5], Raphael Rodriguez [6,7,8], Sanger Mouse Genetics Project, David J. Adams[2] & Stephen P. Jackson[1]

Hutchinson-Gilford Progeria Syndrome (HGPS) is a rare, but devastating genetic disease characterized by segmental premature aging, with cardiovascular disease being the main cause of death. Cells from HGPS patients accumulate progerin, a permanently farnesylated, toxic form of Lamin A, disrupting the nuclear shape and chromatin organization, leading to DNA-damage accumulation and senescence. Therapeutic approaches targeting farnesylation or aiming to reduce progerin levels have provided only partial health improvements. Recently, we identified Remodelin, a small-molecule agent that leads to amelioration of HGPS cellular defects through inhibition of the enzyme N-acetyltransferase 10 (NAT10). Here, we show the preclinical data demonstrating that targeting NAT10 in vivo, either via chemical inhibition or genetic depletion, significantly enhances the healthspan in a *Lmna^G609G* HGPS mouse model. Collectively, the data provided here highlights NAT10 as a potential therapeutic target for HGPS.

[1] The Wellcome Trust/Cancer Research UK Gurdon Institute and Department of Biochemistry, University of Cambridge, Cambridge CB2 1QN, UK. [2] The Wellcome Trust Sanger Institute, Hinxton, Cambridge CB10 1SA, UK. [3] Discovery Sciences, IMED Biotech Unit, AstraZeneca, Cambridge CB4 0WG, UK. [4] Drug Safety and Metabolism, IMED Biotech Unit, AstraZeneca, Cambridge CB2 23AT, UK. [5] Laboratory of Molecular Biology, Cambridge CB2 0QH, UK. [6] Institut Curie, PSL Research University, Paris Cedex 05, France. [7] CNRS UMR3666, 75005 Paris, France. [8] INSERM U1143, 75005 Paris, France. [9] Present address: Department of Clinical Biochemistry, Cambridge Institute for Medical Research, University of Cambridge, Cambridge CB2 0XY, UK. These authors contributed equally: Gabriel Balmus, Delphine Larrieu. Correspondence and requests for materials should be addressed to Sanger.Mouse.Genetics.Project, D.L. (email: dl437@cam.ac.uk) or to S.P.J. (email: s.jackson@gurdon.cam.ac.uk)
A full list of consortium members appears at the end of the paper.

The nuclear envelope (NE) provides a dynamic boundary between the inner nuclear mass and the cytoplasm, and is critical for normal functioning of the eukaryotic cell. Key factors for NE function as a compartmental border are the nuclear lamins, scaffold proteins that specify nuclear architecture and provide mechanical strength to the nucleus and the cell[1]. Notably, over the past few years, lamins have emerged as significant players in many other critical cellular functions, including differentiation, intracellular signaling, chromatin organization, transcription, as well as DNA replication and repair[2,3]. In mammals, the nuclear lamins are categorized in two distinct classes: A-type lamins (Lamin A, Lamin C, Lamin C2, and Lamin AΔ10; all encoded by the *LMNA* gene) and B-type lamins (Lamin B1 encoded by *LMNB1*, and Lamin B2 that, together with Lamin B3, are encoded by the *LMNB2* gene)[4]. In accord with their important roles, the loss-of-function mutations in lamin genes result in genetic syndromes with severe presentations called laminopathies (OMIM #150330; #150340; #150341). These include muscular dystrophies (for example, Emery-Dreyfus Muscular Dystrophy), peripheral neuropathies (for example, Charcot-Marie Tooth-Disease), leukodystrophy, lipodystrophy, as well as premature aging (progeria) syndromes, such as Atypical Werner Syndrome (AWS), Restrictive Dermopathy (RD), and Hutchinson Gilford Progeria Syndrome (HGPS)[2,5]. Of all these syndromes, HGPS is the one with the most striking presentation. After onset, usually within the first year of life, HGPS patients start to display short stature, low body weight (BW), hair loss, lipodystrophy, scleroderma, decreased mobility and osteoporosis, as well as facial features that resemble accelerated aspects of normal ageing[6]. While the cognitive development is normal, cardiovascular abnormalities—characterized by medial smooth-muscle cell loss and secondary maladaptive vascular remodeling (intimal thickening, disrupted elastin fibers, and deposition of atherosclerotic plaques)—are the main reasons for death, with the median life expectancy at birth for HGPS patients being 14.6 years[7–10].

HGPS arises from a heterozygous G608G point mutation of *LMNA* exon 11, leading to cryptic mRNA splicing and expression of a shorter, dysfunctional form of Lamin A, called progerin[11–13]. Similar to wild-type Lamin A, progerin undergoes several post-translational modifications, including the addition of a farnesyl group required for its targeting to the nuclear envelope. However, unlike Lamin A, progerin remains permanently farnesylated, causing it to accumulate on the inner nuclear membrane. In HGPS cells, progerin acts as a dominant-negative protein, aggregating the wild-type lamins, disrupting nuclear shape and chromatin organization, and leading to the increased genomic instability and rapid cell senescence[14,15]. Although farnesyl transferase inhibition (FTI)[16–20] is being explored as a therapeutic approach for HGPS and has provided certain health improvements in patients[21], there is a clear need for additional therapeutic regimes[22].

Recently, we discovered that a small-molecule compound, which we named Remodelin, can ameliorate HGPS cellular phenotypes. Remodelin acts in a progerin- and FTI-independent pathway, by targeting and inhibiting the N-acetyltransferase NAT10[23]. Here, we assess NAT10 inhibition as a potential therapeutic strategy for HGPS by using an established mouse model (*Lmna—G609G* allele) that exhibits premature-aging phenotypes similar to those of HGPS patients[24]. Critical for translating NAT10 inhibition toward human patients, we show that chemical or genetic targeting of NAT10 decreases the genomic instability, and improves age-related phenotypes of both homozygous and heterozygous HGPS mice.

## Results

### Remodelin ameliorates age-dependent weight loss in HGPS mice.
To determine the effects of Remodelin on HGPS mouse cells, we derived skin fibroblasts from *Lmna*^G609G/G609G and wild-type (WT) littermates. As observed in human HGPS fibroblasts[25,26], *Lmna*^G609G/G609G fibroblasts displayed nuclear shape defects and increased genomic instability, as reflected by a higher level of the DNA double-strand break (DSB) marker gamma H2AX (γH2AX, Ser-139 phosphorylated histone H2AX), in a manner that was abrogated by Remodelin treatment (Fig. 1a, b). These results showed that Remodelin treatment can reverse the HGPS induced genomic instability and nuclear shape defects in mouse cells, and provided us with encouraging preliminary evidence to proceed with in vivo studies.

To assess Remodelin's suitability for in vivo studies, we initially defined its pharmacokinetic properties in WT mice. Overall, the oral (PO) delivery appeared to be the best route of administration (Fig. 1c; Supplementary Fig. 1a; Supplementary Tables 1–2), with bioavailability of ~44% (Fig. 1c,) and significant accumulation in heart and skeletal muscle (Fig. 1e). Based on these data, WT and *Lmna*^G609G/G609G mice were treated with a Remodelin dose of 100 mg per kg orally, on a daily schedule from 3 weeks of age onward, until the end-point. This treatment was well-tolerated by both genotypes, with no weight loss (Supplementary Fig. 1b) and with a significant amount of Remodelin being present in the skeletal muscle, liver, and brain of the *Lmna*^G609G/G609G mice (Supplementary Fig. 1c). Furthermore, in line with a previous report that NAT10 promotes melanogenesis[27], we found that Remodelin treatment led to hair graying (Supplementary Fig. 1d). As described before, we found that *Lmna*^G609G/G609G mice had dramatically shorter healthspans, compared to WT controls, associated with accelerated body-weight loss (Fig. 1f; Supplementary Table 3) and premature-aging phenotypes, resembling the human syndrome[24]. Notably, Remodelin led to a 25% increase in Kaplan–Meier area under the curve, based on 20% BW loss, for the treated vs. the vehicle-treated mice (Fig. 1f; Supplementary Table 3). Together, these data established that Remodelin is well-tolerated in vivo and delays the age-dependent weight decline in HGPS mice.

### Remodelin ameliorates cardiac pathology of HGPS mice.
HGPS clinical presentation includes the loss of subcutaneous adipose tissue[7] and cardiovascular abnormalities, such as adventitial fibrosis and medial smooth-muscle cell loss, with depletion of smooth-muscle actin in the remaining cells[8,9,28,29]. These cardiovascular features represent major pathologies that contribute to morbidity and lethality in HGPS[2]. We analyzed all these parameters at the level of the skin, aorta, and coronary heart arteries in Remodelin-treated HGPS mice, as compared to the vehicle-treated controls at their respective end-points. Importantly, Remodelin treatment significantly reduced the loss of subcutaneous adipose tissue that is seen in the HGPS mouse model (Fig. 2a). Moreover, it led to the dramatic amelioration of HGPS cardiac pathologies, including reduction of adventitial fibrosis of the aorta (Fig. 2b), rescue of vascular smooth muscle cell loss (Fig. 2c), and salvage of smooth muscle actin (SMA) loss, both in the aorta and the coronary arteries (Fig. 2d, Supplementary Fig. 2a). By contrast, Remodelin treatment had no significant effect on WT mice at the similar age (Supplementary Fig. 2b). Furthermore, as observed in mouse and human fibroblasts, Remodelin reduced the markers of genome instability in both heart and lung tissues of *Lmna*^G609G/G609G mice (Fig. 2e,f). Together, these data highlighted how Remodelin treatment delayed the onset of cardiovascular pathologies that represent the most debilitating aspect of HGPS.

### Engineering and characterization of a *Nat10* mouse model.
To validate NAT10 as the relevant pharmacological target mediating

the in vivo effects of Remodelin, we engineered a *Nat10* knockout mouse model (Supplementary Fig. 3a). Bi-allelic *Nat10* inactivation was lethal before embryonic day E14.5 (Fig. 3a), indicating that NAT10 is critical for mouse development. However, heterozygous *Nat10*[+/−] mice were born healthy, and were observed to express the *Nat10* mRNA and protein products, at levels ~50% of those in WT animals (Fig. 3b,c).

As NAT10 is largely uncharacterized and has not been studied in mice before, we performed a broad phenotypic analysis of *Nat10*[+/−] mice (Supplementary Fig. 3b, c, Supplementary Fig. 4 and, Supplementary Data 1). In most regards, *Nat10*[+/−] mice were indistinguishable from WT mice (blue color Supplementary Fig. 3b), with the exception of BW, which was consistently lower than that of WT mice, despite similar tail-to-nose lengths associated with changes in the lean and fat mass, triglycerides, and cholesterol levels (Supplementary Fig. 3b, Supplementary Fig. 4a). Additional parameters, including potassium levels and neutrophil numbers, showed evidence of sexual dimorphism

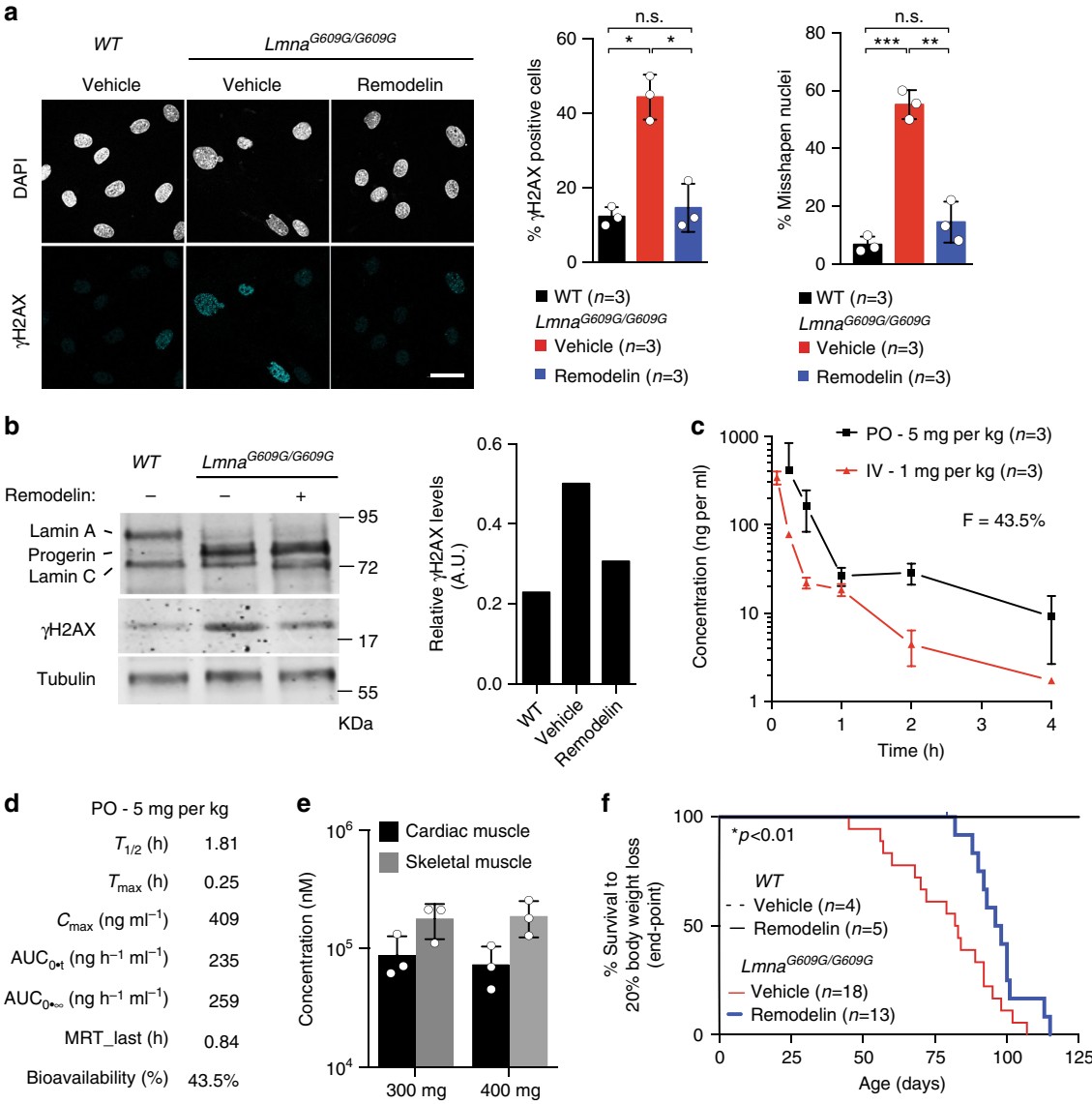

**Fig. 1** Oral administration of Remodelin decreases weight loss in progeria mice. **a, b** Cells were treated with DMSO or with 1 µM Remodelin for 7 days. **a** Left: Representative immunofluorescence images of skin fibroblasts from *Lmna*[G609G/G609G] mice showing the accumulation of the DNA double-strand break marker gamma H2AX (γH2AX) ( blue) and characteristic nuclear shape abnormalities, observed by DAPI staining. All images were acquired with the same microscope intensity settings. Scale bar 20 µm. Right: Quantification of γH2AX positive cells and cells with misshapen nuclei (>100 cells/*n* = 3 independent cell lines; mean ± s.d.; n.s. not significant; *$p < 0.05$, **$p < 0.01$, ***$p < 0.001$, two-tailed Student's *t*-test). **b** Western blotting analysis of γH2AX levels (quantified in the right panel) in skin fibroblasts from the indicated genotypes. **c,d** Pharmacokinetic analyses of Remodelin in mice treated via oral (PO; *n* = 3) or intravenous (IV; *n* = 3) delivery; mean ± s.e.m. F absolute bioavailability (%). **e** Tissues were collected after 2 weeks of daily PO administration of the indicated Remodelin concentration and 1 h after the last dosing. Remodelin was quantified by mass spectrometry in the heart and the skeletal muscle (*n* = 3); mean ± s.e.m. **f** Survival based on 20% body weight loss, showing a 25% increase in Kaplan–Meier area under the curve in Remodelin-treated *Lmna*[G609G/G609G] mice, as compared to vehicle-treated mice (see Supplementary Table 3); (*Log-rank Mantel-Cox test; Chi-square 5.992). Due to animal welfare regulations, mice had to be killed when they had lost 20% of their body weight, compared to their individual weight maxima (end-point). However, at this defined end-point, Remodelin-treated mice displayed considerably better health, compared to the vehicle-treated controls (see supplementary movies and pathology assessments in Fig. 2)

(Supplementary Fig. 4b). Moreover, NAT10 reduction triggered gene-expression changes in the heart (Fig. 3d, Supplementary Data 2), with many deregulated genes being connected to cellular pathways associated with longevity, such as responses to starvation or insulin signaling[30] (Supplementary Fig. 3c). Other highly enriched pathways deregulated upon NAT10 depletion included regulation of inclusion body assembly and protein refolding. These data showed that, while complete NAT10 knockout leads to embryonic lethality, NAT10 haploinsufficient mice

are born at expected frequencies and are overtly healthy, thereby allowing us to explore the potential impacts of reducing *Nat10* gene dosage in the context of a HGPS in vivo genetic model.

**Nat10 haploinsufficiency delays health decline of HGPS mice.** To explore the interaction between NAT10 and HGPS, we generated two cohorts of *Lmna*^G609G/G609G^ mice: one on a WT *Nat10* background (*Nat10*^+/+^) and the other carrying the *Nat10*

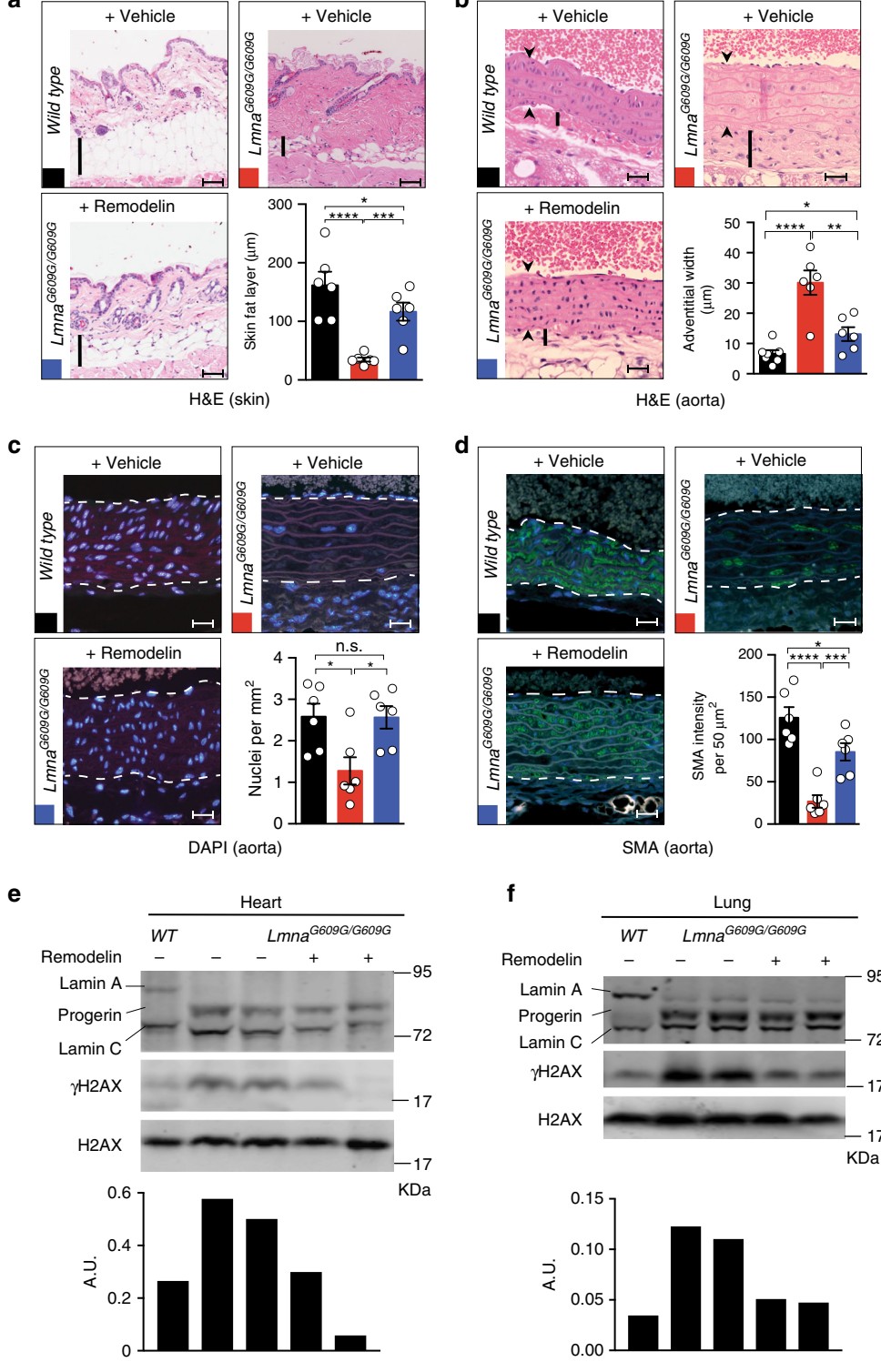

heterozygous deletion ($Nat10^{+/-}$). Notably, as for Remodelin treatment, reduced $Nat10$ gene dosage significantly extended the timeframe of body-weight loss of $Lmna^{G609G/G609G}$ mice by 17% (Fig. 4a; Supplementary Table 3). Moreover, as compared to $Lmna^{G609G/G609G}$ animals, $Lmna^{G609G/G609G}Nat10^{+/-}$ mice displayed a significant delay in the appearance of back curvature (Fig. 4b–d) and enhanced fitness, as observed by lower incidence of common $Lmna^{G609G/G609G}$ mouse pathologies, such as penile prolapse and eye keratitis (Supplementary Fig. 5a). Additionally, $Lmna^{G609G/G609G}Nat10^{+/-}$ mice were more active and overtly healthier than their age-matched controls (Supplementary Movies 1–3).

To better understand how HGPS phenotypes were counter-acted by NAT10 depletion, we performed selected biochemical analyses on mice at 9 and 12 weeks of age. While NAT10 depletion had no effects on parameters such as fat mass and cholesterol levels of progeria mice, it significantly normalized others, including glycerol and urea levels (Supplementary Data 3). Furthermore, the decrease in heart rate observed in $Lmna^{G609G/G609G}$ mice, compared to WT controls, was significantly circumvented by $Nat10$ depletion at 9 weeks, but not at 12 weeks (Fig. 4e), suggesting that the onset of heart abnormalities was delayed. Notably, at 12 weeks, $p21$ expression was increased in the hearts of $Lmna^{G609G/G609G}$ mice but not in $Lmna^{G609G/G609G}Nat10^{+/-}$ mice, probably reflecting higher DNA-damage loads (Fig. 4f).

While $Lmna^{G609G/G609G}$ mice were reported to be infertile[24], under our husbandry they were sub-fertile: $Lmna^{G609G/G609G}$ males had decreased sperm counts (Supplementary Fig. 5b), but these sperms were motile and able to fertilize wild-type oocytes (Supplementary Fig. 5c). Similarly, $Lmna^{G609G/G609G}$ females produced morphologically normal oocytes that were meiotically competent (Supplementary Fig. 5d, e), and the super-ovulated eggs from $Lmna^{G609G/G609G}$ females could be fertilized by WT sperm (Supplementary Fig. 5c). However, while $Lmna^{G609G/G609G}$ females never produced litters, $Lmna^{G609G/G609G}$ males could father pups, albeit at a very low frequency and never more than one litter. These data suggested that sub-fertility of HGPS mice was caused by decreased reproductive fitness, rather than by a specific physiological problem. Strikingly, global fitness/health improvements associated with reduced $Nat10$ gene dosage in $Lmna^{G609G/G609G}Nat10^{+/-}$ mice (Supplementary Movies 2–3) correlated with an enhancement of male and female fertility (Supplementary Fig. 5f; 45% in $Lmna^{G609G/G609G}Nat10^{+/-}$ vs. 21% in $Lmna^{G609G/G609G}$).

While we, like others, have used homozygous $Lmna^{G609G/G609G}$ to model HGPS, it is important to note that patients carry a heterozygous $Lmna$ mutation, leading to the expression of both WT Lamin A and progerin. We therefore wished to study the effect of reducing $Nat10$ gene dosage in heterozygous $Lmna^{+/G609G}$ mice. Extending upon the previous report showing

reduced lifespan of such mice[24], we performed an extensive phenotypic analysis of $Lmna^{+/G609G}$ mice and found them to display similar premature aging phenotypes as the homozygous mutant (Supplementary Fig. 6, Supplementary Data 4), albeit with delayed onset. When we assessed the impact of $Nat10$ heterozygosity in this human-disease model, we found that $Lmna^{+/G609G}Nat10^{+/-}$ mice lived significantly longer before displaying age dependent body-weight decline, compared to $Lmna^{+/G609G}Nat10^{+/+}$ mice. Indeed, there was a 90 day gap between the longest lived (based on 20% BW loss) $Lmna^{+/G609G}Nat10^{+/-}$ mouse and the longest lived $Lmna^{+/G609G}Nat10^{+/+}$ mouse (Fig. 4g; Supplementary Table 3). In addition, the $Lmna^{+/G609G}Nat10^{+/-}$ mice displayed decreased back curvature (Supplementary Fig. 7). Taken together, these data show that $Nat10$ haploinsufficiency enhances health in both the homozygous and the heterozygous HGPS mouse models.

**Identification of readouts for NAT10 inhibition in HGPS.** To investigate the impacts of NAT10 inhibition in vivo, and as the main cause of death in HGPS patients is due to heart dysfunction, we performed global gene-expression analyses on tissues derived from hearts of $Lmna^{G609G/G609G}Nat10^{+/-}$ and Remodelin-treated $Lmna^{G609G/G609G}$ mice, compared to their respective controls (Supplementary Data 5). This work identified a specific set of genes (Supplementary Fig. 8a)—largely connected to metabolic pathways (Supplementary Fig. 8b)—as significantly upregulated (blue) or downregulated (red) in $Lmna^{G609G/G609G}$ mice, compared to WT (Supplementary Fig. 8a, lane 1). Interestingly, some of these gene-expression differences (* in Supplementary Fig. 8a) were counteracted by both Remodelin treatment (row 2) and $Nat10$ depletion (row 3). Further investigation would be required to establish whether these might comprise a gene-expression signature for amelioration of HGPS pathologies by NAT10 inhibition. Collectively, these data highlighted a strong correlation between NAT10 chemical inhibition and its genetic depletion on cellular imbalances, caused by the $Lmna$ G609G/HGPS mutation. They also suggested the potential for gene-expression signatures as readouts for HGPS and its alleviation.

As acetylation of α-tubulin lysine 40 (K40) is a documented NAT10 target[31,32], we evaluated it as another potential readout for NAT0 inhibition in vivo. Indeed, quantitative analyses established that patient-derived HGPS fibroblasts, as well as $Lmna^{G609G/G609G}$ mouse-derived skin cells and tissues, displayed increased acetyl-α-tubulin K40, when compared to WT controls (Fig. 5; Supplementary Fig. 9), probably associated with increased microtubule stability, contributing to HGPS cellular phenotypes[25]. However, this was not associated with significant and consistent increased NAT10 protein levels, suggesting that NAT10 enzymatic activity might be elevated, leading to increased α-tubulin acetylation. In line with these findings, Remodelin

**Fig. 2** Remodelin ameliorates cardiac and other pathologies of HGPS mice. Pathological staining in panels **a–d** was carried out on the materials from end-point mice (presented in Fig. 1f) of indicated genotypes ($n = 6$ per genotype) treated with vehicle or Remodelin 100 mg per kg per day (for detailed ages of the mice, see Supplementary Table 4). All images are representative (scale bar 50 μm) and the correspondent bar graph quantifications are presented (mean ± s.e.m.; individual data points represented; n.s. not significant; *$p < 0.05$, **$p < 0.01$, ***$p < 0.001$, ****$p < 0.0001$ two-tailed Student's t-test). In WT mice, Remodelin treatment has no significant effect, as compared to vehicle treatment; and for simplicity these animals have been pooled in one group; the individual comparison is presented in Supplementary Fig. 2b. **a** Hematoxilin and eosin (H&E) staining of skin, indicating fat layer thickness (vertical bars indicate the fat layer) and showing amelioration of the fat layer thickness upon Remodelin treatment in HGPS mice. **b** H&E staining of heart aorta, indicating increased adventitial width in the HGPS mice, as compared to WT controls, which is rescued by Remodelin treatment (arrowheads demarcate the aorta and vertical bars the adventitia). **c** DAPI staining of heart aorta, showing a decreased number of nuclei in the HGPS mice, rescued by Remodelin treatment (dotted white lines delineate the aorta edges). **d** Smooth muscle actin (SMA) staining (green) of heart aorta sections, showing loss of integrity of the artery wall in HGPS mice, improved by Remodelin treatment (dotted white lines delineate the aorta). **e, f** Representative western blotting analysis of the representative heart (**e**) and lung (**f**) tissues from end-point mice, showing that Remodelin decreased γH2AX levels in $Lmna^{G609G/G609G}$ tissues (see quantification below each Western blot, relative to total H2AX levels). Western blots were performed more than once on $n \geq 4$/group

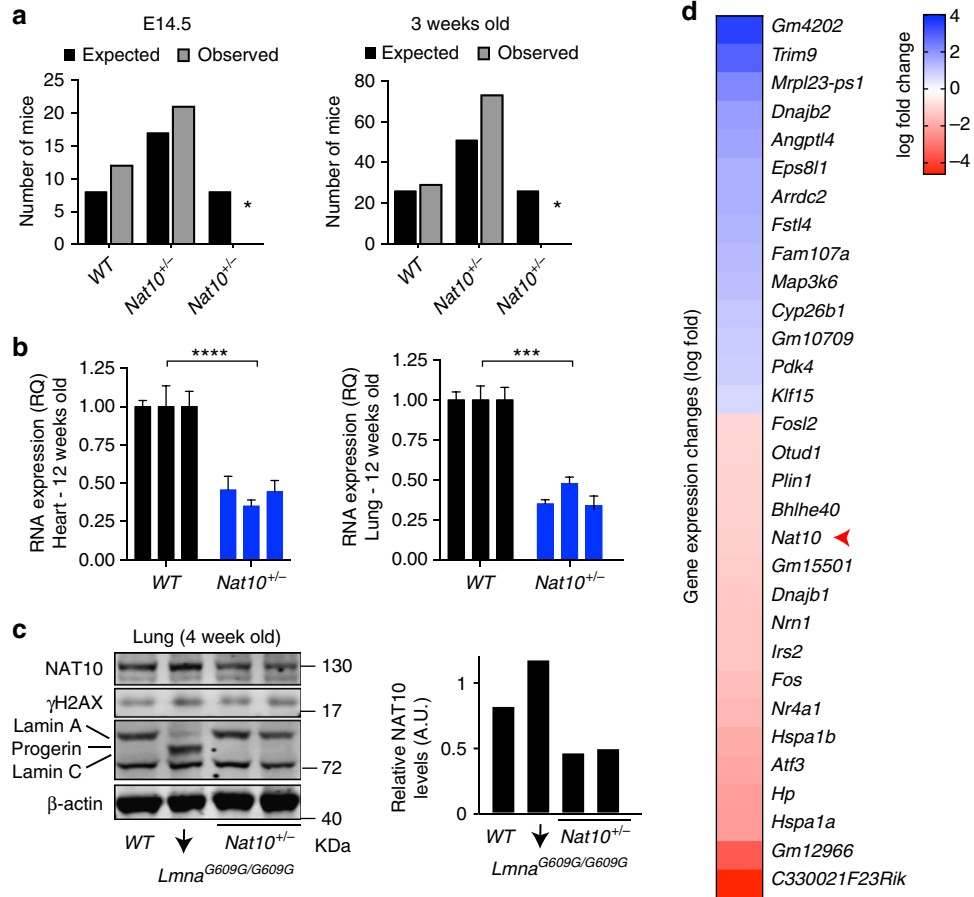

**Fig. 3** Engineering and characterization of a Nat10$^{+/-}$ mouse model. **a** Number of observed embryos (E14.5) and mice (21 days), compared to the expected numbers (Mendelian frequencies) (*$p < 0.01$; Chi-square analysis). **b**, **c** Nat10$^{+/-}$ mice display ~50% reduction in Nat10 transcript level (**b** each bar indicates individual mice ± s.d. of $n = 5$ technical replicates/mouse) or protein expression in the indicated tissues (**c** representative western blot; blots have been performed more than once on $n \geq 3$ mice), and the quantification is presented on the right panel, relative to actin levels. **d** Heatmap of genes differentially expressed in the heart of Nat10$^{+/-}$ mice, compared to the wild-type, from RNAseq analysis ($n = 2$/genotype)

treatment or reduced Nat10 gene dosage decreased acetyl-α-tubulin K40 levels in extracts from cells and mouse tissues (Fig. 5a, b; Supplementary Fig. 9a, b), in cultured cells (Fig. 5c), and as detected by in situ mouse-tissue staining (Fig. 5d; Supplementary Fig. 9cd). These results thus indicated that α-tubulin K40 acetylation is increased in both human HGPS cells and in the mouse HGPS model, in a manner counteracted by Remodelin treatment, thereby suggesting this acetylation mark as a potential readout for NAT10 inhibition in vivo.

## Discussion

HGPS, a debilitating premature aging disease whose features strikingly resemble the accelerated aspects of normal aging, represents a paradigm for translational medicine in the arena of aging research[33]. The complex nature of this segmental syndrome makes it difficult to target therapeutically, with standard therapeutic approaches mainly aiming at preventing the expression or the accumulation of progerin on the nuclear envelope. Thus, strategies that have so far received most attention largely involve the targeting enzymes participating in the progerin pathway[10], including HMG-CoA reductase, farnesyl-pyrophosphate synthase, farnesyl transferase[21], isoprenylcysteine carboxyl methyl-transferase, as well as modulating Lmna pre-mRNA splicing by morpholino compounds[24]. Notably, there is encouraging evidence that HGPS children treated with the FTI lonafarnib display

improved vascular stiffness and bone structure[21], as well as 33% increased survival, based on Kaplan–Meier area under the curve estimations[10]. Because FTI treatment is now essentially standard-of-care and the HGPS patient population is small (~1 in 18 million)[34], further clinical trials will likely be arranged as combined therapies with lonafarnib[22].

Our results show that NAT10 genetic depletion or its chemical inhibition by the compound Remodelin lead to healthspan enhancements— as indicated by effects on age-dependent BW loss, cardiac function, back curvature, fitness, and genomic instability—in both homozygous and heterozygous HGPS mice, through a mechanism that appears to be independent of progerin. Reinforcing our conclusions, we have recently carried out further studies with a Remodelin derivative, Remodelin-fluor (Supplementary Fig. 10a), that in HGPS cells showed the same effect as Remodelin at half the Remodelin dose (Supplementary Fig. 10b). Using this compound in mice, we observed hair graying (Supplementary Fig. 10c) and no drug-induced weight loss (Supplementary Fig. 10d). Moreover, Remodelin-fluor treatment led to a significant decrease in the timeframe of body-weight loss along with 30% increase in Kaplan–Meier area under the curve for treated mice, as compared to vehicle-treated mice (Supplementary Fig. 10e; Supplementary Table 3). While we did not see Remodelin-induced effects on low mineral density and bone mineral content in HGPS mice (see Supplementary Data 3), we observed significant disease amelioration by Remodelin treatment

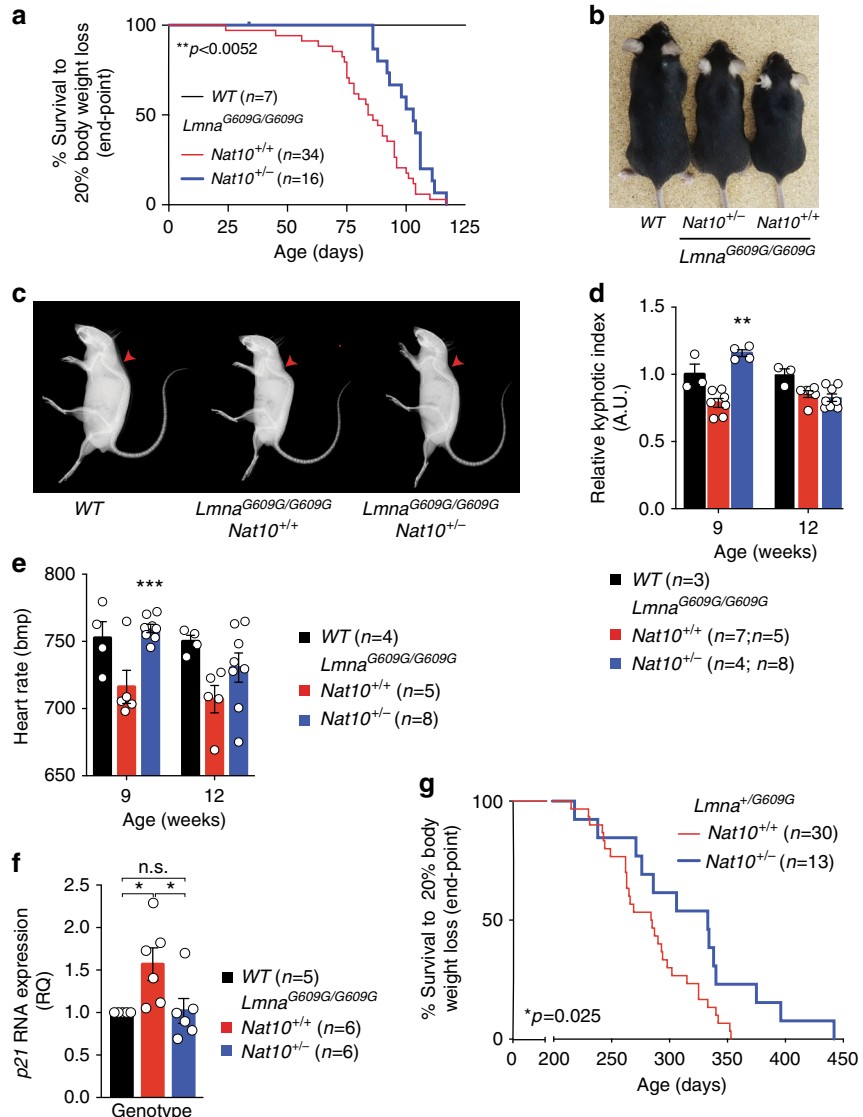

**Fig. 4** Genetic depletion of *Nat10* enhances the health of *Lmna*[G609G] mice. **a** *Lmna*[G609G/G609G]*Nat10*[+/−] mice show a 21% increased median age at the end-point, compared to *Lmna*[G609G/G609G] (103 days vs. 85 days respective median age at end-point; based on the mice being terminated upon reaching 20% body weight loss); (**Log-rank, Mantel-Cox test; Chi-square 32.61; also see Supplementary 3). **b–d** Appearance (**b**) of back curvature (**c**) in *Lmna*[G609G/G609G] mice is delayed by *Nat10* depletion, as observed by images of terminal mice and X-rays from 9 week-old females, and quantified (kyphotic index) over time (**d**) (mean ± s.e.m.; individual data points represented; mixed model analysis shows a significant difference between *Lmna*[G609G/G609G]*Nat10*[+/+] and *Lmna*[G609G/G609G]*Nat10*[+/−] genotypes **p = 0.01; raw data and extended conclusions and statistics are presented in Supplementary Data 3). **e** Progressive heart function failure observed in *Lmna*[G609G/G609G] mice over time is delayed by *Nat10* depletion, as observed by the heart rate measurements at indicated times (mean ± s.e.m.; individual data points represented; mixed model analysis shows a significant difference between *Lmna*[G609G/G609G]*Nat10*[+/+] and *Lmna*[G609G/G609G]*Nat10*[+/−] genotypes ***p = 0.004; raw data, extended conclusions, and statistics are presented in Supplementary Data 3). **f** RNA expression from heart tissues shows decreased *p21* expression in *Lmna*[G609G/G609G]*Nat10*[+/−], compared to controls (mean ± s.e.m.; n.s. not significant, *p < 0.05 two-tailed Student's *t*-test; individual data points represented). **g** *Lmna*[+/G609G]*Nat10*[+/−] mice show a 17% increased median age at end-point, compared to *Lmna*[+/G609G] (333 days vs. 285 days respective median age at end-point; based on mice being terminated upon reaching a 20% body weight loss); (*Log-rank, Mantel-Cox test; Chi-square 4.98; also see Supplementary Table 3) and more than 90 days between the longest lived (20% body weight loss) *Lmna*[+/G609G]*Nat10*[+/−] and the longest lived *Lmna*[+/G609G] mouse

and *Nat10* genetic depletion in critical organs, such as the heart. Notably, these cardiovascular effects were not only restricted to the aorta, but also included impacts on other large vessels of the heart, such as the coronary arteries. These effects are of potential clinical relevance because advanced coronary disease is prevalent in HGPS patients, even in the absence of high blood pressure[29,35]. We also note that, in contrast to FTI[36], Remodelin decreases markers of genome instability in HGPS cells and organs. We showed in our previous work that combining FTI and Remodelin

in HGPS patient cells does not improve the cellular phenotypes further, compared to Remodelin alone (see Supplementary Fig. 8 of Larrieu et al., Science 2014[25]). However, as they act in different pathways, it is possible that the phenotypes in vivo would benefit from the drug combination, which will likely be the scope of ensuing studies.

Collectively, our data thus highlights the potential for NAT10 inhibitors in treating HGPS in combination with lonafarnib, where additive therapeutic effects might be anticipated. While we

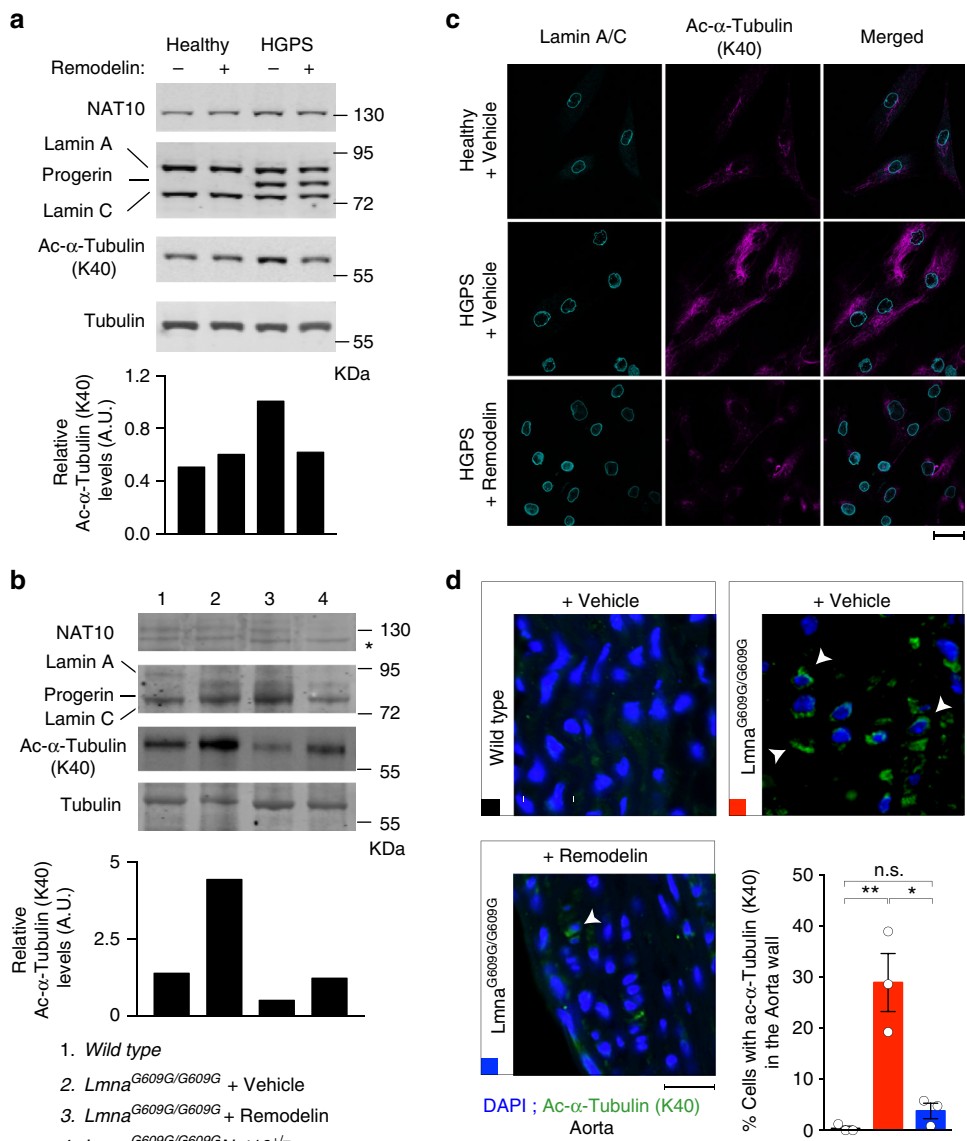

**Fig. 5** Identification of potential readouts for *Nat10* inhibition in cells and tissues. **a,b** Representative images of western blots showing that 1 μM Remodelin treatment for 7 days decreases the high alpha-Tubulin (α-tubulin) K40 acetylation in HGPS-patient derived cells (**a**) and mouse tissues (**b**). In panel **b** NAT10 chemical (lane 3) or genetic (lane 4) inhibition reverses the high α-tubulin K40 acetylation levels in heart tissues from indicated mice; *indicates a cross-reacting band. We note that the ratio between Lamin A and C appears to vary between tissues. All western blotting experiments were performed independently at least three times (n ≥ 3/genotype). **c** Representative immunofluorescence images of acetyl-α-tubulin K40 in HGPS-patient derived cells, as compared to matching healthy fibroblasts. Scale bar = 20 μm. The K40 α-tubulin acetylation (magenta) is increased in the HGPS-patient derived cells and decreased upon Remodelin treatment. **d** Representative immunofluorescence images (left) and quantification (right) of acetyl-α-tubulin K40 in aortas of terminal mice of the indicated genotypes and treatments. Scale bar = 10 μm. The K40 α-tubulin acetylation (green; white arrowheads point to example of cells that show increased K40 acetylation) is increased in *Lmna*^G609G/G609G mice and significantly decreased in such mice, upon Remodelin treatment (n = 3; mean ± s.d.; individual data points represented; n.s. not significant, *p < 0.05; **p < 0.01; ***p < 0.001; two-tailed Student's *t*-test). For better visualization, these are higher magnification snapshots (red dotted squares) from images in Supplementary Fig. 9d. Quantification was performed on full size aorta images from n = 3 independent mice

found that complete *Nat10* deletion resulted in early embryonic lethality in mice, *Nat10* haploinsufficiency or chemical inhibition via Remodelin treatment did not confer any profound side effects. Because NAT10 is a 115 kDa protein with multiple functional domains, it could be that the lethality associated with its total loss is linked to another aspect of the NAT10 protein, other than its N-acetyltransferase function. In this regard, we note that in *Caenorhabditis elegans*, a null allele of NAT10, *nath-10(tm2624)*, causes fully penetrant embryonic lethality in the homozygous state. By contrast, the *nath-10(N2)* hypomorphic allele,

containing a polymorphism in the N-acetyltransferase domain, did not cause pathology in a homozygous setting, but instead conferred increased fitness and a strong competitive advantage over WT animals[37]. While these and our data are encouraging from the perspective of considering NAT10 inhibition as a therapeutic approach for HGPS, the fact that our understanding of NAT10 biology is still in its infancy means that any clinical studies should be explored with caution.

The small number of HGPS patients and the diverse nature of their pathologies provide challenges for the evaluation of

potentially new HGPS therapies in the clinic[22]. If NAT10 inhibitors are explored in clinical settings, it will thus be extremely valuable to assess target engagement. In this regard, it will be of interest to investigate the potential of the gene-expression signatures that we have found to be associated with HGPS cells, and which are rebalanced toward WT profiles by either Remodelin treatment or NAT10 depletion. Furthermore, our data suggests that acetylated α-tubulin lysine 40 (K40), a known NAT10 target[31,32], might also be used as a readout for NAT10 inhibition in cells and in vivo. These findings also correlated with our previous data, indicating that NAT10 inhibition ameliorates HGPS cellular phenotypes, at least in part, by mediating microtubule destabilization[25]. As α-tubulin acetylation at K40 is elevated in HGPS tissues and the cells are compared to controls, it will be of interest to explore whether this could be used to monitor disease progression and also enhance our understanding of disease pathobiology. Finally, we speculate that because the hallmarks of HGPS are present at lower levels in the vasculature and other tissues of aged-normal individuals[33], NAT10 targeting might offer therapeutic opportunities in broader settings. In accord with such a possibility, we have recently reported the effects of NAT10 inhibition in normally aged smooth muscle cells[26], that might suggest the potential for NAT10 inhibition in the context of normal ageing.

## Methods

**Synthesis of Remodelin and derivatives**. All solvents and reagents were purified using standard techniques, or used as supplies from commercial sources (Sigma-Aldrich). NMR spectra were acquired on a Bruker 500 MHz instrument, with deuterated solvents at 300 K. Notation for the [1]H NMR spectral splitting patterns includes: singlet (s), doublet (d), triplet (t), broad (br), and multiplet/overlapping peaks (m). Signals are quoted as values in ppm and coupling constants (J) are quoted in Hertz. Mass spectra were recorded on a Micromass® Q-Tof (ESI) spectrometer. General procedure: The appropriate ketone or aldehyde was dissolved in isopropanol at a final concentration of 0.5 M and refluxed for 24 h in the presence of an equimolar amount of thiosemicarbazide. The corresponding thiosemicarbazones were isolated by filtration and recrystallized from hot ethanol. Equimolar amounts of thiosemicarbazones and the desired haloketones were stirred at room temperature in isopropanol overnight at a final concentration of 0.2 M. The resulting products were recrystallized from hot ethanol several times to yield pure products and were used without further purification.

4-(4-cyanophenyl)-2-(2-cyclopentylidenehydrazinyl)thiazole (Remodelin): 2-cyclopentylidenehydrazine-1-carbothioamide (1 g, 4.46 mmol) and 2-bromo-4'-cyanoacetophenone (700 mg, 4.45 mmol) were stirred overnight in 12 ml of isopropanol at room temperature. The precipitate was filtered and recrystallized from hot ethanol to yield the hydrobromide salt of the desired compound (559 mg, 1.98 mmol, 45%) as light-yellow needles.[1]H NMR (500 MHz, CDCl₃): δ 12.11 (br s), 7.84 (d, J = 9.0 Hz, 2H), 7.81 (d, J = 9.0 Hz, 2H), 6.84 (s, 1H), 2.61 (t, J = 9.0 Hz, 2H), 2.51 (t, J = 9.0 Hz, 2H), 1.94–1.80 (m, 4H); [13]C NMR (125 MHz, CDCl₃): δ 173.8, 169.5, 138.8, 133.5, 131.3, 126.3, 118.0, 114.1, 103.8, 33.7, 31.2, 25.2, 25.0; HRMS (m/z): [M]⁺ calcd. for $C_{15}H_{15}N_4S$, 283.1009; found, 283.1017. Molecule **4** was resuspended in DMSO at 10 mg/ml.

4-(4-trifluoromethylphenyl)-2-(2-cyclopentylidenehydrazinyl)thiazole (Remodelin Fluor): 2-cyclopentylidenehydrazine-1-carbothioamide (3.0 g, 18.7 mmol) and 2-bromo-4'-(trifluoromethyl)acetophenone (5.0 g, 18.7 mmol) were stirred in isopropanol (150 mL) at r.t. for 24 h. The precipitate was filtered and recrystallized three times from hot ethanol to yield the hydrobromide salt of Remodelin Fluor as bright yellow needles (1.5 g, 3.7 mmol, 20%). [1]H NMR (300 MHz, CDCl3) δ 11.12 (br s, 2H), 7.80 (d, J = 8.5 Hz, 2H), 7.65 (d, J = 8.5 Hz, 2H), 6.87 (s, 1 H), 2.53 (t, J = 7.0 Hz, 2 H), 2.47 (t, J = 7.0 Hz, 2 H), 1.93–1.75 (m, 4 H). 13 C NMR (75 MHz, CDCl3) δ 171.9, 169.4, 140.3, 131.8 (q, J = 32.0 Hz), 131.6, 126.5 (br q, J = 3.5 Hz, 2 C), 126.1 (2 C), 123.7 (q, J = 274.0 Hz), 103.5, 33.5, 30.6, 25.1, 24.9. HRMS (m/z): [M + H] + calculated for C15H15F3N3S, 326.0933; found, 326.0950. See Fig. S2A for the synthesis reaction.

**Animals—ethical information**. For the studies at the Wellcome Trust Sanger Institute (WTSI) the care and use of all mice used to generate data for this protocol was carried out in accordance with UK Home Office regulations, UK Animals (Scientific Procedures) Act of 2013 under UK Home Office licenses which approved this, and were reviewed regularly by the WTSI Animal Welfare and Ethical Review Board. For the studies at Crown Biosciences the protocols and any amendment(s) or procedures involving the care and use of animals in this study were reviewed and approved by the Institutional Animal Care and Use Committee (IACUC) of Crown Biosciences. During the study, the care and use of animals were conducted in accordance with the regulations of the Association for Assessment and Accreditation of Laboratory Animal Care (AAALAC AN-1308-017-66).

**Animals—housing and husbandry**. At WTSI, mice are maintained in a specific pathogen-free unit on a 12-h light and 12-h dark cycle with lights off at 19:30 and no twilight period. The ambient temperature is 21 ± 2 °C, and the humidity is 55 ± 10%. Mice are housed using a stocking density of 3–5 mice per cage (overall dimensions of caging: 365 × 207 × 140 mm (length × width × height), floor area 530 cm²) in individually ventilated caging (Tecniplast, Sealsafe 1284 L) receiving 60 air changes per hour. In addition to Aspen bedding substrate, standard environmental enrichment of two Nestlets, a cardboard fun tunnel and three wooden chew blocks are provided. Mice were given water and diet (Teklad Global 18% Protein Rodent Diet/Envigo) ad libitum. At Crown Biosciences, mice were housed at an average temperature of 23.5 °C with a 7:00 am–19:00 pm light and 19:00 pm – 7:00 am (next day) darkness cycle in polysulfone IVC cages (3 mice/cage; 325 mm × 210 mm × 180 mm). Mice were fed Co⁶⁰ irradiation sterilized dry granule mouse diet. Animals had free access to food and water during the entire study period. Mice had no drug or test naïve prior to treatment.

**Engineering of the *Nat10^tm1a(KOMP)Wtsi* (*Nat10^+/−*) mice**. Mice carrying the knockout-first conditional-ready allele *Nat10^tm1a(KOMP)Wtsi* (abbreviated to *Nat10^tm1a*) were generated on a C57BL/6 N background as part of the Sanger Mouse Genetics Project (MGP). Detailed description of the Sanger Mouse Genetics Project methodology has been reported[38]. Briefly, a promoter-containing cassette (L1L2_ Bact_P) was introduced upstream of the critical *Nat10* exon 4 at position 103754720 of Chromosome 2, Build GRCm38. The vector containing *Nat10^tm1a* was electroporated into C57BL/6 N derived JM8A3.N1 ES cells. Correct ES cell gene targeting was confirmed by long-range PCR and quantitative PCR. Targeted ES cells were microinjected into blastocysts and used to generate chimeras. Germline transmission was confirmed by genotyping PCR analyzes (http://www.knockoutmouse.org/kb/25/). Mice obtained from heterozygous intercross were genotyped for the *Nat10^tm1a* allele by PCR. Mice were killed by CO₂ inhalation followed by cervical dislocation.

**Use of the *Lmna^G609G* and *Lmna^G609G* *Nat10^+/−* mice**. *Lmna^G609G* (C57BL/6N) mice were imported from the laboratory of Carlos-Lopez Otin[39] and re-derived on the line C57BL/6NTac at the Wellcome Trust Sanger Institute (WTSI). Mouse genotyping was performed from tail biopsies[39]. Nat10 KO mice were generated in the C57BL/6NTac background as part of the Mouse Genetics Project at the WTSI[40]. The double mutant combinations were maintained on the same C57BL/6NTac background. Experimental animals were maintained under close supervision according to the following protocol. Upon weaning *Lmna^G609G/G609G* homozygous animals and littermate *Lmna^+/+* wild-type (WT) controls were set up in experimental cohorts and weighed weekly. WT and mutant experimental animals were housed together randomly in multiple cages to reduce cage bias. Treatment and weight measurements were carried out by the mouse facility technicians blind of the scientific background or goals of the experiment. When animals reached 10% BW loss they were weighed every other day and wet pellets provided on the floor daily. Upon 15% BW loss animals were weighed and monitored daily and culled when they passed over 19% BW loss. A number of male mice of the *Lmna^G609G/G609G* genotype have presented with penile prolapse and upon detection they have been culled and indicated as incidence of penile prolapse (Supplementary Fig. 5a). Over the course of the study three mice of the *Lmna^G609G/G609G* genotype have been found dead. No drug or naïve test was performed prior to treatment and testing. Animals have been killed by CO₂ inhalation followed by cervical dislocation. WTSI facility runs periodic health reports that indicate that the mice were free of known viral, bacterial and parasitic pathogens. For analysis of embryonic development of Nat10 KO mice, timed matings were performed at noon and the day of vaginal plug detection was defined as embryonic day E0.5. Movies and pictures were made using a Sony Cyber-shot DSC-HX10V GPS camera. Mouse health evaluation was performed by trained technicians using established protocols[41].

**Small-molecule dosing**. Remodelin and Remodelin Fluor were dissolved in a solution of 20% DMSO, 65% (45% 2-Hydroxypropyl-b-cyclodextrin solution, H5784 Sigma Aldrich) and 15% Tween 80 (P8074 Sigma Aldrich). The vehicle-treated animals were given this solution alone, without Remodelin. Remodelin and Remodelin Fluor were administered daily by oral gavage at 100 mg per kg per day and 50 mg per kg per day respectively (defined as non-toxic doses in toxicity studies), from day 21 and until culled. Dosing needle—Instech FTP-20–30 Plastic feeding tubes, 20ga × 30 mm 1 ml syringes and 20ga dosing needles were used as they are appropriate for the volume to be administered and for the size of the mice. During each dosing session, the vehicle only was administered to each animal in the control group before administering Remodelin to each animal in the treated group to avoid cross-contamination. BWs were recorded for each mouse before dosing and the dose volume was calculated according to the BW. The mice were restrained by scruffing the back of the neck, the dosing needle was presented through the mouth down into the esophagus in a smooth motion. Once in situ, the dose (calculated to a 100–200 μl volume) was administered to the mouse.

Additional food was moistened and added to the floor of each cage to facilitate food consumption following the dosing procedure. The end-point criteria were represented by 20% BW loss, mice that were found dead or moribund or mice that presented with penile prolapse, a distinctive clinical phenotype for the progeric male mice. All animals were used to represent the survival curves; $n = 1$ Remodelin-treated $Lmna^{G609G/G609G}$ mouse presented with haemangioma at 79 days of age and was censored into the survival analysis. Treatment and weight measurements were carried out by the mouse facility technicians blind of the scientific background or goals of the experiment. The comparison between $Lmna^{G609G/G609G}$ vehicle and $Lmna^{G609G/G609G}$ no treatment showed no significant difference ($p = 0.56$). We utilized an informal method of randomizing within each batch to assign mice to treatment using mouse database ID. Mice were used from multiple cages and litters.

**Assessment of remodelin toxicity in vivo and dosing.** Remodelin toxicity was assessed by the Crown Biosciences on 12 6-week-old C57BL/6N female mice, with a BW of 20 g on average (see Table S2. Animal supplier: Shanghai Laboratory Animal Center SLAC, Shanghai, China). At WTSI, Remodelin and Remodelin Fluor were dissolved in a solution of 20% DMSO, 65% (45% 2-Hydroxypropyl-b-cyclodextrin solution, H5784 Sigma Aldrich) and 15% Tween 80 (P8074 Sigma Aldrich). The vehicle-treated animals were given this solution alone, without Remodelin. Remodelin and Remodelin Fluor were administered daily by oral gavage at 100 mg per kg per day and 50 mg per kg per day respectively (defined as non-toxic doses in toxicity studies), from day 21 and until culled. Animals have been killed by $CO_2$ inhalation followed by cervical dislocation.

**Pharmacokinetics evaluation of Remodelin.** The pharmacokinetics of Remodelin was assessed by the Crown Bioscience (Supplementary Tables 1 and 2) and the Xenogesis Ltd., Nottingham, UK (Supplementary Fig. 1c). The administration of Remodelin and sample collection in each WT study group are shown in the following experimental design tables Tables S1 and S2 (IV = intravenous; PO = per os (by mouth)). The $Lmna^{G609G}$ mice were treated with 100 mg/kg Remodelin for 6 weeks and tissues and plasma collected 1 h after the last dosing. Animals were randomly assigned to groups. Before grouping and treatment, all animals were weighed, and assigned into groups using randomized block based on their BW. Within each block, experimental animals were randomly assigned to the different groups. Randomized block design was used to assign experimental animals to ensure that each animal has the same probability of being assigned to any given treatment groups and therefore minimizing systematic error. Animals showing obvious signs of severe distress and/or pain were humanely killed. Animal that had lost significant body mass (>20%, emaciated) were killed. Animals found to have other severe health problems, i.e., prolonged diarrhea, persistent anorexia, lethargy or failure to respond to gentle stimuli, labored respiration, or that could not get to adequate food or water, etc., were removed from the study and killed ($n = 3$; subcutaneous administration; data not shown). LC/MS/MS method development was used to assess the test compound in plasma. A minimum of 6 standards with LLOQ < 5 ng/mL and a minimum of five standards back were calculated to within ±20% of their nominal concentrations. A total of six QC samples at three concentrations (Low, Mid, and High) were included in sample runs with a minimum of four QC back were calculated to within ±20% of their nominal concentrations. Plasma samples were analyzed following the above criteria.

**Cell lines.** Normal skin primary fibroblasts GM03440 and Hutchinson Gilford Progeria Syndrome (HGPS) skin primary fibroblasts AG11513 and A11498 were purchased from Coriell Cell Repositories and used between passage number 9–17. Cells were grown in Dulbecco's modified Eagle medium (DMEM, Sigma-Aldrich) supplemented with 10% fetal bovine serum (BioSera), 2mM L-glutamine, 100 U/ml penicillin, 100 μg/ml streptomycin. All cell lines were tested for mycoplasma contamination using the Charles River Mycoplasma Testing Services.

**Antibodies.** Antibodies used in this study are: Lamin A/C (sc-6215 Santa-Cruz 1:200 for Western Blotting and 1:100 for Immunofluorescence), NAT10 (13365-1-AP ProteinTech Europe 1:400 for Western Blotting), γH2AX (05-636 Millipore, 1:200 for Western Blotting and 1:100 for Immunofluorescence), H2AX (ab11175 Abcam, 1:500 for Western Blotting), β Actin (ab8226, Abcam, 1:1000 for Western Blotting), Acetyl α Tubulin (K40) (5335 Cell Signaling, 1:500 for Western Blotting and 1:100 for Immunofluorescence), α Tubulin (T9026 Sigma-Aldrich, 1:1000 for Western Blotting and 1:400 for Immunofluorescence).

**Immunoblotting of mouse tissues.** Mice were killed by $CO_2$ inhalation followed by cervical dislocation. Mouse tissues were snap-frozen in liquid $N_2$ and immediately stored at −80 °C. 40 μl/g of protein extraction buffer (50 mM Tris pH 7.5, 150 mM NaCl, 0.1% NP-40, 0.5% CHAPS, 5 mM $MgCl_2$, 10% glycerol, MilliQ distilled water) or Laemmli Buffer (S3401 SIGMA). Mini-protease and mini-phosphatase tablets were added to each sample in conjunction with a stainless-steel bead (7 mm; QIAGEN) and physically disrupted using the TissueLyser LT (QIAGEN), in 2 × 5 min cycles with 5 min rest on ice in between. After removing the stainless-steel bead, the lysates were sonicated using the BioruptorTM Next Gen (Diagenode), 2 × 20 min cycles (30 s on, 30 s off), and a 20 min rest period on ice in

between. The tissue debris was separated from the protein content by centrifugation using a bench Eppendorf centrifuge 5417R (Eppendorf) for 30 min, at 4 °C, and 16400 rpm. The quantity of protein was measured using Pierce$^{TM}$ BCA protein assay kit (Thermo Scientific) and Multiskan GO (Thermo Scientific) at an absorbance of 562 nm.

**Immunoblotting of cell extracts.** Total cell extracts were prepared by scraping cells in SDS lysis buffer (4% SDS, 20% glycerol, and 120 mM Tris-HCl, pH 6.8), boiling for 5 min at 95 °C, followed by ten strokes through a 25-gauge needle. Before loading, cell or tissue lysates were diluted with a solution of 0.01% bromophenol blue and 200 mM DTT and boiled for 10 min at 95 °C. Proteins from individual mice/cell lines were resolved by SDS-PAGE on 4%–12% gradient gels (NUPAGE, Life Sciences) and transferred onto nitrocellulose membranes (Protran; Whatman). Secondary antibodies conjugated to IRDye were from LI-COR Biosciences. Western blots shown are representative of three repeats. Detection and quantification was performed with an imager (Odyssey; LI-COR Biosciences). Uncropped Western blots are presented in Supplementary Figure 11.

**Immunofluorescence.** Isolation and culture of adult mouse fibroblasts from skin and lungs were performed using an established protocol[42]. Cells were washed with PBS and fixed for 10 min with 4% PFA in PBS. Cells were permeabilised for 5 min with PBS/0.2% Triton X-100 and blocked with PBS/0.2% Tween 20 (PBS-T) containing 5% BSA. Coverslips were incubated for 1 h with primary antibodies and for 30 min with appropriate secondary antibodies coupled to Alexa Fluor 488 or 594 fluorophores (Life Technologies), before being incubated with 2 μg/ml DAPI. Pictures were acquired with a FluoView 1000 confocal microscope (Olympus) and images were quantified using ImageJ. All the immunofluorescence experiments were performed independently at least three times and the pictures shown in the figures are representative images of at least three experiments.

**Histology and immunohistochemistry.** All major organs were isolated following killing and then fixed in 10% formalin overnight. On the second day the fixed organs were transferred to 70% ethanol, were placed in cassettes, embedded in paraffin and serial 5 μm sections were collected on Superfrost Plus slides (Fisher) using a Leica microdissection system (LMD7000). Hematoxilin and eosin (HE) and immunohistochemistry staining were performed using standard protocols[43]. The organs were examined for abnormalities by a Board Certified Veterinary Pathologist (MJ). Sections of heart containing aortic outflow, pulmonary artery and myocardium were stained using a primary antibody against alpha smooth muscle actin (SMA-Sigma Aldrich Cat No: A2547 1:1000) and a secondary antibody (Alexa 488 Life Technologies at 1:100). The heart sections were also stained using DAPI alone and with an antibody against Acetyl α Tubulin (K40) (5335 Cell Signaling). Representative histology images were obtained from whole slide images scanned on a Hamamatsu NanoZoomer in brightfield and fluorescence modes. The thickness of the subcutaneous fat layer in the skin and nuclei number in the aorta were measured using whole slide images and the Hamamatsu NDP view software at a magnification of ×200 (resolution of 0.45 μm per pixel). The skin (always the same region/the flanks) and aorta regions for analysis were identified by a pathologist (MJ) and manually annotated using the HALO image analysis software (Indica Labs). The cells inside these annotated regions were identified and counted using the HALO software CytoNuclear v1.5 algorithm, the output of cell density (cells/mm$^2$) was used to differentiate between the treatment groups. SMA intensity quantification was performed using ImageJ.

**Oocyte culture and immunofluorescence.** Oocytes were isolated from ovaries of 8-week old C57BL/6 N female mice and cultured in M2 medium covered by mineral oil at 37 °C. Isolated oocytes were maintained in prophase arrest by addition of 250 μM dbcAMP (Sigma; D0627). To induce resumption of meiosis, oocytes were released from dbcAMP-free medium. For immunofluorescence, oocytes were fixed for 30 min at 37 °C in 100 mM HEPES (pH 7; titrated with KOH), 50 mM EGTA (pH 7; titrated with KOH), 2% formaldehyde (methanol free) and 0.2% Triton X-100. Fixed oocytes were incubated in PBS with 0.1% Triton X-100 overnight at 4 °C. Antibody incubations were performed in PBS, 3% BSA, and 0.1% Triton X-100. Primary antibodies used were rat anti-Nup98 (Abcam, ab50610; 1:100) and rat anti-tyrosinated-α-tubulin (YOL1/34, AbD Serotec; 1:3000). Secondary antibodies used were Alexa-Fluor-488-labeled anti-rat (Molecular Probes; 1:400). DNA was stained with 5 mg/ml Hoechst 33342 (Molecular Probes). Images were acquired with a Zeiss LSM710 microscope equipped with a 63 × C-Apochromat 1.2 NA water-immersion objective.

**Generation of oocytes and in vitro fertilization.** Three 4- to 5-week-old C57BL/6NTac females per sperm sample were super-ovulated by intraperitoneal (IP) injection of 5IU of pregnant mare's serum at 17:00 h (on a 12 h light/dark cycle, on at 07:00/off at 19:00) followed 48 h later by an IP injection of 5IU human chorionic gonadotrophin. Oviducts were dissected at ~07:50 am on the day of the in vitro fertilization (IVF), and cumulus-oocyte complexes were transferred into the IVF fertilization dish containing human tubal fluid (HTF) + glutathione (GSH). An aliquot of 20 μl of sperm from the pre-incubation dish was then added to the

fertilization dish. After allowing 3–4 h for fertilization to occur the embryos were washed and cultured overnight in HTF at 37 °C, 5% $CO_2$ in air.

**High-throughput phenotypic screen**. The high-throughput phenotyping screen, is a series of standardized tests performed on all mice that enter the screen and conducted according to standard operating procedures (SOPs). The tests cover a broad range of biological areas, including metabolism, cardiovascular, neurological and behavioral, bone, sensory and hematological systems and plasma chemistry. The data were obtained following the SOPs available at IMPReSS (www.mousephenotype.org/impress)[40]. Factors predicted to affect the variables were standardized where possible. If this was not possible, measures were taken to reduce potential bias. For example, the impact of different people performing the test was minimized (known as the "minimized operator") as defined in the Mouse Experimental Design Ontology (MEDO)[44]. The data captured with the MEDO ontology can be found at http://www.mousephenotype.org/about-impc/arrive-guidelines. In addition, pre-established reasons were defined for QC failures (e.g., insufficient sample, error with equipment during test) and detailed using IMPRESS. This provides standardized options and criteria as agreed by area experts as to when data can be discarded. All discarded data is retained and tracked in a database to allow QC-failed data to be audited. Phenotyping data are collected at regular intervals on age, sex, and strain-matched wildtype (control) mice. In total, at least seven homozygote mice of each sex per knockout line were generated for phenotyping. If no homozygotes were obtained from ≥28 offspring from hetero-zygote intercrosses at P14, the line was declared homozygous lethal. Similarly, if less than 13% of the pups resulting from intercrossing were homozygous survive to P14, the line was judged as being homozygous subviable. In this event, heterozygote mice were examined in the phenotyping screen. The random allocation of mice to experimental group (wildtype vs. knockout) was driven by Mendelian Inheritance. Because of the high-throughput nature of the phenotyping screen, blinding the operators to the identity of knockout lines (both which line to be studied and the zygosity of the individual mouse) during phenotyping was not employed as the cage cards used to identify the mice includes genotype information. However, in a high throughput environment without a defined hypothesis, the potential bias is minimized. In all cases, the individual mouse was considered the experimental unit. Further experimental design strategies (e.g., exact definition of a control animal) is defined using a standardized ontology as detailed in Karp et al.[44] and is available from the IMPC portal (http://www.mousephenotype.org/about-impc/arrive-guidelines). For each line, $n \geq 5$ mice/genotype were studied.

**Comparison between mouse genotypes**. Mutant mice at 9 or 12 weeks of age were analyzed in separate groups ($n \geq 3$ for each sex/genotype). Prior to entering this workflow, due to the progeria phenotype, mice were assessed for adverse health and welfare to ensure that mice were in a suitable condition for analysis. No mice were excluded from study on this basis. Mice were anaesthetized with 110 mg/kg BW ketamine and 11 mg/kg BW xylazine given intraperitoneally. Mice were then imaged sequentially with three modalities, high resolution X-Ray imaging (MX-20, Faxitron, Tucson, AZ), Dual-energy X-ray Absorptiometry for body composition (Piximus II, GE Healthcare, Hatfield, UK), and with light photography for imaging of dysmorphology. A whole-body lateral image using the MX-20 was collected for the analysis of spinal curvature by trained persons using a standard defined position to minimize inconsistencies between mice. Following this, and while still under anesthesia, blood collection was performed to obtain samples for plasma chemistry and hematological analysis *via* the retro-orbital route using capillary tubes (cat. no. 078042; Scientific Laboratory Supplies). Mice were then culled by cervical dislocation and heart removal, followed by removal of other organs for analysis. During this series of procedures, the experimenters collecting images and blood samples were not blinded from the genotypes of the mice. However, the analysis of the blood parameters was performed blind and uploaded onto a data-base. The kyphosis index (KY) was calculated as the ratio between a line drawn between the caudal margin of the last cervical vertebra to the caudal margin of the sixth lumbar vertebra and a line perpendicular to this from the dorsal edge of the vertebra at the point of greatest curvature[45]. For the data collected on the back curvature, a multilevel regression model was performed using R (package:nlme version 3.1). A model (Eq. 1), treating genotype, sex and age as fixed effects whilst the repeat measure nature of the dataset was accounted for by treating each mouse as a random effect, was fitted to the data. The genotype effect was tested and contrasts used to directly compare $Lmna^{G609G/G609G}$ and $Lmna^{G609G/G609G}Nat10$ $^{+/-}$ mice if the genotype effect was significant. For the hypothesis test of primary interest, the impact of genotype, p-values were adjusted to account for the multiple comparisons completed to control the false discovery rate to 0.05. Visual inspection of the data was used to assess whether variance was equal and no outliers were present thus ensuring the assumptions of the model were met.

$$Y \sim Genotype + Sex + Age + (1|Mouse) \quad (1)$$

For the survival analysis, the end-point criteria were represented by 20% BW loss, mice that were found dead or moribund or mice that presented with penile prolapse, a distinctive clinical phenotype for the progeric male mice. All animals were used to represent the survival curves; one $Lmna^{G609G/G609G}Nat10^{+/-}$ was reported by the technician staff with a swollen abdomen at 32 days of age and

requested to be culled. The mouse was active and within the weight range for the age. No abnormality was found at necropsy but because it was culled for other reasons than end-point criteria it was censored from the analysis.

**Statistical analysis**. For all analyzes, the individual mouse was considered the experimental unit within the studies. Survival distributions of the different cohorts were plotted using the Kaplan–Meier estimator and statistical analysis was per-formed using log-rank (Mantel–Cox) test. For survival analysis, we have completed power calculations for a large size effect and with an n of 10/group. We could detect a 0% to 60% change in survival after treatment, 91% of the time (power of Fisher Exact test, 0.91). To meet the assumption of this statistical method, cen-soring of an individual mouse could only occur when the culling of a mouse was not related to the genotype/assessed-phenotype (e.g., fight wound leading to overt clinical presentation).

**Nat10$^{+/-}$ and Lmna$^{G609G}$ high-throughput phenotyping data**. Knockout data collected across multiple batches were compared to a year's worth of control data collected on mice from the same genetic background. For the continuous data, an iterative top down mixed modeling strategy fitting Eq. 1 was performed using PhenStat[46], an R package version 2.6.0[47] freely available from Bioconductor[48]. The package's mixed model framework was used as default except the argument equationType was set to withoutWeight and dataPointsThreshold was set to 2. The model optimization implemented will adjust for unequal variances. The genotype contribution test p value was adjusted for multiple testing to control the false discovery rate to 5%. This statistical method has been studied through simulations and resampling studies[49] and found to be robust and reliable with a multi-batch workflow, where the knockout mice are split into multiple phenotyping batches.

$$Y \sim Genotype + Sex + Genotype * Sex + (1|Batch) \quad (2)$$

For the categorical data, a Fisher Exact Test was fitted comparing the proportions seen between wildtype and knockout mice for each sex independently using PhenStat FE Framework with the default settings. This simple method is appropriate for categorical phenotyping data as discussed in Karp et al.[46]. The minimum p value returned from the two tests for a variable was adjusted for multiple testing to control the false discovery rate to 5%. The number of caudal vertebrae were recoded to a categorical variable by classifying mice with less than 28 vertebrae as "low," those with greater than 29 as "high" and all others as "normal." For ABR data, the knockout dataset was smaller with only four data points for each variable, therefore to meet the assumption of the test, the data was analyzed using a reference range plus methodology which calls a significant phenotype when the majority of animals lie outside the natural variation seen in the control animals[47]. The implementation within PhenStat RR framework is based on classifying the analyzable variable values as high, normal or low based on the natural variation seen within the control data and comparing the proportions seen with a Fisher Exact Test. The minimum p value returned from the two tests ((1) increase in high classification and (2) increase in low classification) for a variable was adjusted for multiple testing to control the false discovery rate to 5%. As a high throughput program with many variables and multiple analysis tools, a single power calculation would not help; instead, the pipeline has been developed through empirically selecting a workflow which has historically given hits at a rate that would be cost effective for the program.

**Data analysis for genotype comparison**. Mixed model data analysis was per-formed using R (package: nlme version 3.1). An iterative top down modeling strategy was implemented starting with the fully loaded model (Eq. 2). For the Origins of Bone and Cartilage Disease (OBCD) screen[50], where only one sex was collected, Eq. 3 details the starting model. The final model was selected by first selecting a structure for the random effects, then a covariance structure for the residual, and then the model reduced by removing non-significant fixed effects. Then the genotype effect was tested, model diagnostics assessed and contrasts used to directly compare $Lmna^{G609G/G609G}$ and $Lmna^{G609G/G609G}Nat10^{+/-}$ mice if the genotype effect was significant. During the model building stage, the hypotheses were tested with a threshold of $p < 0.05$. For the hypothesis test of primary interest, the impact of genotype, p-values were adjusted to account for the multiple com-parisons completed to control the false discovery rate to 0.05. The difficulty associated with the breeding and viability challenged the production of these mice for the array of phenotyping used in this paper. 136 mating pairs were set up over more than 3 years that generated 181 $Lmna^{G609G}$ homozygous (single or $Nat10$ double-mutant) mice that were used at specific ages together with littermate controls. The n was thus limited by breeding constraint and we have used $n > 5$.

$$Y \sim Genotype + Age + (1|Batch) \quad (3)$$

**Heart rate comparison**. Heart measurements were performed using the ecgTUN-NEL (emka TECHNOLOGIES), a noninvasive ECG system, using the manufacturer's recommendations[51]. All ECG recording sessions were performed during daytime and the data analyzed using the *iox2*, data acquisition, and analysis software (emka

TECHNOLOGIES). Each animal was put inside the tunnel, which was then closed, ensuring the animal was properly restrained. To minimize the effects of stress, animals were allowed to stay in the restraining system for 1 min before starting ECG recordings. Indeed, direct observation of the animals and ECG traces proved that they were calm and that the heart rate was stable. For data acquisition, a series of repeated measurements were done on the same animal at each time-point and data for the same animal was collected over the different week intervals. For the data collected on the heart rate screen, a multilevel regression model was performed using R (package: nlme version 3.1). A model [Eq. 1], treating genotype, sex, and age as fixed effects whilst the repeat measure nature of the dataset was accounted for by treating each mouse as a random effect, was fitted to the data. The genotype effect was tested and contrasts used to directly compare $Lmna^{G609G/G609G}$ and $Lmna^{G609G/G609G}Nat10^{+/-}$ mice if the genotype effect was significant. For the hypothesis test of primary interest, the impact of genotype, p-values were adjusted to account for the multiple comparisons completed to control the false discovery rate to 0.05. Visual inspection of the data was used to assess whether variance was equal and no outliers were present thus ensuring the assumptions of the model were met.

**RNA extraction and qPCR analysis**. RNA was extracted from tissues from $n > 5$ independent mice/group using the RNeasy fibrous tissue mini kit (50; cat. No.7404; Qiagen) and quantified using the NanoDrop 1000 Spectrophotometer (Thermo Fisher Scientific). 2 µg RNA/sample was used to produce cDNA using the High-Capacity RNA-to-cDNA kit (cat. No. 4387406; Applied Biosystems/Thermo Fisher Scientific). qPCR was carried out using the TaqMan system (Universal Master Mix II, with UNG, 4440038; Applied Biosystems/Thermo Fisher Scientific) on an Applied Biosystems Quant Studio 3 machine. $n \geq 3$ mice were used for each genotype, with 50 ng cDNA for each sample run in triplicate or quadruplicate. Thermo Fisher Scientific Nat10 (Mm00462302_m1) and Cdkn1a/p21 (Mm00432448_m1) primers were used as experimental probes while Gapdh (Mm99999915_g1), Actb (Mm00607939_s1) and/or Rn18s (Mm03928990_g1) were used as endogenous controls. The Relative Standard Curve pre-set program was used throughout and data analysis was performed using the Relative Quantification application, powered by the Thermo Fisher Scientific cloud platform. To account for technical replicates the mean value for a mouse was calculated and used for the statistical assessment. For each sample, data was normalized to the endogenous controls and represented as relative to the wildtype control. Data was evaluated by visual inspection and an F test of the variance was calculated in order to estimate normality and equal variance. All graphs and part of the statistical analysis in the manuscript (Student's t-tests; Kaplan–Meier estimator and statistical analysis; Fishers exact test) were generated and calculated using GraphPad Prism version 7.0a for Mac OS X, GraphPad Software, La Jolla, California, USA, www.graphpad.com.

**RNAseq data analysis**. RNA was extracted as described above and quality control assessed using the 2100 Bioanalyzer (Agilent Technologies). Because of financial constrains we used $n = 2$ mice/group. Transcriptome data was obtained using paired end sequencing, with read lengths of 150 bp, on a NextSeq 500 machine. Trimmed reads were aligned using STAR aligner (version 2.4.2a) to the mouse genome assembly GRCm38. Normalization of the read counts and differential expression analysis was performed using three commonly used software programs: DeSeq2[52], edgeR[53], and Cuffdiff 2[54]. Our conservative approach defined genes differentially expressed as those found to be in common between the results of at least two of the three software programs mentioned above. The log-2 fold change in gene expression presented in Fig. 2d and Fig. 4e correspond to the log-2 fold change returned by DeSeq2. DeSeq2 and edgeR used as an input raw read counts produced by featureCounts from the Bioconductor (version 3.3) Rsubread package (14) in R version 3.3.1. For the DeSeq2 and edgeR analyzes we filtered out genes that had 0 or 1 read support across all samples. In the DeSeq2 differential expression analysis, we selected genes that were up or down regulated at a FDR lower than 0.1. In edgeR differential expression analysis, we selected genes up or down regulated with a p-value less than 0.05. Gene ontology analysis was performed using the mouse genome informatics visual annotation display[55]. For the analysis of the biological term fold enrichment, a ratio between the frequency of genes in our set to frequency of genes in the whole genome was calculated[56].

**Data availability**. All data presented in the manuscript are available from the corresponding authors upon reasonable request. The mouse phenotypic data from the present manuscript are available in the supplementary Data. Mouse phenotypic data are available from IMPC and Zenodo. RNAseq data are available from ArrayExpress under accession code E-MTAB-6578.

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

## Acknowledgements

We thank all members of the Steve Jackson laboratory for help and support—particularly Kate Dry for editing the manuscript—and Dr Carlos Lopez-Otin for sharing his *Lmna*$^{G609G}$ mouse model with us. Research in the Jackson laboratory is funded by Cancer Research UK (CRUK) program grant C6/ A18796 and a Wellcome Trust Investigator Award (206388/Z/17/Z). Institute core funding is provided by CRUK (C6946/A24843) and the Wellcome Trust (WT203144). D.L was funded by a Project Grant from the Medical Research Council, UK MR/L019116/1. Research in the D.J.A. laboratory is supported by CRUK and the Wellcome Trust. Research in the R. R. laboratory is supported by the European Research Council (Grant N° 647973), and the Emergence Ville de Paris Program. M.D. was supported by the European Research Council grant DDREAM. Some data in this publication form part of the subject matter of patent WO 2015/150824. The funders had no role in study design, data collection, and analysis, decision to publish or in preparation of the manuscript.

## Author contributions

G.B., D.L., and S.P.J. coordinated and conceptualized the study and wrote the manuscript. D.L. performed the human and mouse derived cell culture, prepared the small-molecule doses for animal gavage, generated immunofluorescence and western blots with help from MD and analyzed the related data. G.B. analyzed the survival and fertility data, performed the heart rate measurements, kyphotic index measurements, and sperm counts as well as mouse primary cell line derivations. G.B. helped with tissue collection throughout the study. A.B. genotyped all the mice in this study, did all the mouse protein extractions and analyzed the weighting data. C.C. did the Remodelin and Remodelin fluor gavages and mouse weighting with covering help from other mouse facility staff and supervised daily by A.K. A.K. helped throughout the study with end-point criteria assessment making sure consistency was achieved. M.A. performed the RNAseq analysis and pathway analysis with help from G.B. N.G. performed the RNA extractions and did the qPCR analysis with help from G.B. C.J.L. and J.K.W. supervised the phenotypic analysis; C.J.L. assembled all the phenotypic pipeline raw data produced by Sanger Mouse Genetics Project that performed all the pipeline phenotypic measurements in this study as well as the IVF analysis. N.K. performed the statistical analysis on the phenotypic pipeline data and helped throughout with all the statistical analysis in the paper. N.K. made an important contribution to manuscript material and methods writing. M.J. and J. A. performed the pathologic staining and assessments. D.C. performed the meiotic spreads analysis. R.R. designed and synthesized the small molecules Remodelin and Remodelin Fluor used in the study. D.J.A. helped supervise all the mouse work. S.P.J. and D.L. supervised the work. All the authors commented and edited the manuscript and figures.

## Additional information

**Competing interests:** D.L., S.P.J., and R.R. are named inventors on a patent describing compounds that include Remodelin. The remaining authors declare no competing interests.

## Sanger Mouse Genetics Project

Carl Shannon, Mark Sanderson, Amy Gates, Joshua Dench, Valerie Vancollie, Catherine McCarthy, Selina Pearson, Emma Cambridge, Christopher Isherwood, Heather Wilson, Evelyn Grau, Antonella Galli, Yvette E. Hooks, Catherine L. Tudor, Angela L. Green, Fiona L. Kussy, Elizabeth J. Tuck, Emma J. Siragher,

Robbie S.B. McLaren, Agnieszka Swiatkowska, Susana S. Caetano, Cecilia Icoresi Mazzeo, Monika H. Dabrowska, Simon A. Maguire, David T. Lafont, Lauren F.E. Anthony, Maksymilian T. Sumowski, James Bussell, Caroline Sinclair, Ellen Brown, Brendan Doe, Hannah Wardle-Jones, Nicola Griggs, Mike Woods, Helen Kundi, George McConnell, Joanne Doran, Mark N.D. Griffiths, Christian Kipp, Simon A. Holroyd, David J. Gannon, Rafael Alcantara, Ramiro Ramirez–Solis, Joanna Bottomley, Catherine Ingle, Victoria Ross, Daniel Barrett, Debarati Sethi, Diane Gleeson, Jonathan Burvill, Radka Platte, Edward Ryder, Elodie Sins, Evelina Miklejewska, Dominique Von Schiller, Graham Duddy, Jana Urbanova, Katharina Boroviak, Maria Imran & Shalini Kamu Reddy

