## [Peer Review File · Nature Communications]

Reviewers' comments:

Reviewer #1 (Remarks to the Author):

The manuscript from the Jackson lab describes a *in vivo* treatment of a progeria syndrome including a mechanism that works via inhibition of N-acetyltransferase 10.

This is an important finding with translational potential. A mechanism has been described. I have a few points the authors should take into account:

- Throughout the manuscript, figures are falsely referenced.
- For many data it remains unclear whether technical or biological replicates had been used, in particular because sample numbers seem somewhat low in some experiments.
- Representative Western blots should come with corresponding quantifications of all replicates and statistical evaluation, in particular when proposing a western-based acetylation signal as a potential biomarker.
- Since the increase in tubulin-acetylation in HGPS patients/models can not be causally linked to NAT10 via changes in protein levels or enzymatic activity, the value of tubulin-acetylation as a readout for NAT10 function remains questionable. Please discuss this carefully.

Reviewer #2 (Remarks to the Author):

Balmus et al. report lifespan extension and physiological benefits in a mouse model of the accelerated aging disease Hutchinson-Guilford progeria syndrome by treatment with a small molecule called remodelin, a previously reported putative inhibitor of the protein acetyltransferase NAT10. Remodelin treatment of an established mouse model of HGPS (G609G) leads to a 25% lifespan extension and improvement of cardiac and adipose phenotype. To validate the involvement of NAT10 in HGPS phenotype, the authors generate a NAT10 knockout mouse. Nat10-nulls are embryonic lethal and Nat10 heterozygotes have reduced body weight, altered lipid composition, and hints of sexual dimorphism. Crosses of Nat10^{+/-} with HGPS(G609G) animals showed rescue of several HGPS phenotypes including back curvature, penile relapse, eye keratitis and some cardiac features. Similar effects were observed when G609G-heterozygous mice were used in the crosses. The authors also perform gene expression analysis on G609G-remodelin and G609G/Nat10 animals and identify a set of genes which they suggest may be as biomarkers. Finally, the authors provide evidence for α -tubulin K40 acetylation as a biomarker in HGPS.

The most interesting finding in this study is the demonstration of rescue of some disease symptoms after reduction of NAT10 levels. The effect of remodelin in the HGPS model will be of interest to the field. However, several conclusions are not convincingly supported by the presented data and are largely speculative, there are experimental shortcomings, and critical controls are missing.

The two major weaknesses of this study are:

The first major weakness of the study is that it is not clear whether the observed effect of remodelin is specific to HGPS or represents a general beneficial effect. The longevity study on WT animals was not carried out long enough to assess this critical question. Ideally, the authors would need to test remodelin in a separate accelerated aging model to answer this important question.

The second major weakness is that the authors indicate that due to animal welfare regulations, animals had to be sacrificed when they had lost 20% of their body weight. These data are presented in figure 1e and 4a but are presented as "survival" data. Although I appreciate the requirement to follow animal welfare policy, the conclusions are presented are highly misleading. The data should be labelled as "time of 20% weight loss". More importantly, it is not appropriate to

conclude that these data can be used as an indicator of survival. The authors assume that body weight necessarily tracks with survival, but do not provide any data to support this critical assumption. Furthermore, the authors speculate that the lifespan effect would be even larger if measured directly, but we simply do not know and no data to support this extrapolation is provided. No conclusions, including in the title and abstract, regarding lifespan extension should be made. This is particularly true for the Nat10/HGPS cross which only showed a moderate 17% shift in 20% body weight loss.

Additional points in order of priority:

The quality of data in figure 5a, b is very poor and the presented conclusions are not convincingly supported by these data:

- All western data should be quantified and normalized to tubulin levels or another cellular protein.
- Multiple WT and HGPS cell lines should be tested.
- In contrast to the authors' statement there appears to be more NAT10 in HGPS cells than controls.
- The increase in α -tubulin K40ac is not evident in figure 5a.
- The NAT10 blot in figure 5b is very poor.
- Why is there more lamin C than lamin A in the WT in figure 5b?
- Why is there no progerin in G609G/Nat10 \pm animals (figure 5b, lane 4).

Does remodelin have an effect on normal aging or is its effect limited to HGPS?

The significance of the gene expression profiles used to identify potential biomarkers is not clear. These proposed markers are not validated and not directly tested. This is a speculative conclusion.

Figure 1: The authors show rescue of γ H2AX by remodelin treatment in G609G mice. DNA damage is one of many nuclear defects. The authors should test for rescue of some of the other prominent nuclear defects.

p. 10. Top of page, the reference should be to figure 4g not 5g.

Reviewer #3 (Remarks to the Author):

Comments to the Authors:

This manuscript by Gabriel Balmus et al. reports in vivo studies of a new Nat10 inhibitor Remodelin in LMNAG609G HGPS mice and in LMNAG609G mice with Nat10 partial depletion. The HGPS mice with either Remodelin treatment or partial Nat10 knockdown showed beneficial effects by extending mice lifespan and improving their cardiovascular pathology and function with increased genomic integration reflected by γ H2AX reduction. Those effects via targeting Nat10 are independent of disease-causing progerin expression. As the first-time mouse study for Remodelin, the authors conducted the toxicity evaluation and provided pharmacokinetics profile in WT mouse. They further studied its derivative and found Remodelin-fluor is more effective and less toxic to mice.

Although this study is of great interest in exploring a novel, potential therapeutic remedy for HGPS patients, additional experiments need to be performed to strengthen the main conclusions. Some parts of this manuscript should be described clearly with detailed information and the whole writing should proofread carefully to avoid any confusion to readers. Outlined below are specific

comments to help improve this manuscript.

Major concerns:

1. Skin fibroblasts derived from LMNAG609G/G609G mouse showed less percentage of γ H2AX positive cells upon Remodelin treatment compared to vehicle-treated cells in Figure 1a. The image quality is low and needs to be improved. There is no description for the scale bar in figure legend. Detailed information for how long the cells were treated at what concentration of Remodelin should clearly indicate in figure legend or Method (same for Figure 5a-5b). The quantification of γ H2AX-positively stained cells collected by manual counting in histogram is likely subjective based on how to define "Positive cells". Western blotting analysis for γ H2AX expression level normalized to total H2AX should be more accurate to provide quantitative results as shown in Figure 2e.

2. Which administration PO or IV was performed for calculating pharmacokinetic (PK) parameters presented in Figure 1c? The description in corresponding figure legend is quite confusing. Please clarify. How long after PO administration of high dose Remodelin were the tissues collected for mass spectrometry assay in Figure 1d?

3. PK studies usually show gender-based differences. Are there any specific reasons to use mice with different gender in two sets of PK studies in Table S1 and S2? There is no PK result in the entire manuscript that is corresponding to Table S2 and it is very important to check Remodelin concentrations in heart and muscle after PO or IV dosing at 100 mg/kg. Have the authors conducted PK study in LMNAG609G/G609G HGPS mice and checked Remodelin concentrations in their main organs, including heart/aorta, liver, kidney, lung, brain, bone, and skin? The accumulation of Remodelin after multiple or long-term administrations in different tissue and organ would probably provide some explanations for their pathological phenotype changes.

4. The description for Remodelin toxicity assessment is confusing. If I understand correctly, the authors conducted a sub-acute toxicity trial (Figure S1a) after repeated Remodelin administration by PO or SC for two-week duration, and a chronic toxicity trial by PO dosing Remodelin at 100 mg/kg/day for 13 weeks (Figure S1b) and Remodlin-fluor at 50 mg/kg/day. They claimed that this new chemical at 100 mg/kg/day is well tolerated without body weight loss except "the only observed side effect being hair greying". Since it is a brand new chemical, further study for its toxicity is highly suggested, e.g. acute toxicity assessment to determine LD50 (Median Lethal Dose) and MTD (Maximum Tolerated Dose) that will help optimize the ideal dosage for lifespan trial, the histopathology examination in main organs like liver, kidney, lung, etc. in chronic toxicity trial, etc.

5. Authors showed pathological improvement in G609G mouse skin and aorta after Remodelin treatment in Figure 2a at endpoint of individual mouse. It's confusing. Please clarify the specific endpoint for each mouse in Figure 2a-2d. Normally, such comparison should be conducted at same time point after drug treatment.

6. Figure 2e: Please explain why there was no visible lamin A band in heart tissues from G609G mouse with/without Remodelin treatment, and why the lamin C in two vehicle-treated samples had a clear band shifting compared to WT and treated sample. If this is due to some technical issues, please provide better representative WB data.

7. Authors claim the molecular mechanism underlying Remodelin effects partly through deacetylation of α -tubulin (Figure 5 and Page 14 Paragraph 2): "Remodelin treatment decreases the high α -tubulin K40 acetylation in HGPS-patient derived cells (a) and mouse tissues (b)." and "... indicating that NAT10 inhibition ameliorates HGPS cellular phenotypes, at least in part by mediating microtubule destabilization." How to explain the elevated level of acetylated α -tubulin at lys-40 in HGPS cells and mouse tissues? Nat10 expression actually looks higher in HGPS fibroblasts compared to Healthy control (Figure 5a), quantitative analysis should be provided to

compare its expression between disease and control cells. What's Nat10 activity in disease cells and tissues vs. control? What's the functional correlation of α -tubulin K40 acetylation to nuclear morphology?

8. Authors state (Page 12): "... Remodelin leads to lifespan and fitness improvements in both homozygous and heterozygous HGPS mice via a mechanism that appears to be independent of progerin." Previous study from same group also indicates Remodelin-induced improvement of the nuclear shape in HGPS cells without reduction of progerin expression. Are there any changes for progerin distribution in HGPS nucleus after Remodelin treatment? The redistribution of progerin from nuclear rim into nucleoplasm could improve nuclear morphology due to less tethering of progerin to nuclear envelope. Immunofluorescence staining of progerin protein in HGPS cells upon Remodelin treatment would seem adequate to exclude this situation.

Minor Concerns:

There are many wrong references in the entire manuscript including supplementary materials. Please proofread carefully and rearrange or correct them.

For example:

- 1) Supplementary Material Page 7: The title is "Pharmacokinetics Evaluation ...". It is actually more about lifespan study design (Table S3, Paragraph 198~199) but not for PK study (Table S1 & S2). The description for MS assay (Page 6 last paragraph) was put in toxicity session.
- 2) Supplementary Material Page 23 Figure S2: "Identification of a more potent Remodelin analogue". However, this S2 is about the coronary pathology.

Reviewer #1 (Remarks to the Author):

The manuscript from the Jackson lab describes a in vivo treatment of a progeria syndrome including a mechanism that works via inhibition of N-acetyltransferase 10. This is an important finding with translational potential. A mechanism has been described. I have a few points the authors should take into account:

We thank Reviewer 1 for his/her positive comments.

-Throughout the manuscript, figures are falsely referenced.

We apologize for this, and have rectified the errors in our revised manuscript.

- For many data it remains unclear whether technical or biological replicates had been used, in particular because sample numbers seem somewhat low in some experiments.

These have now been corrected/updated; please note that for the majority of the data, the individual points represent individual mice (thus biological replicates).

- Representative Western blots should come with corresponding quantifications of all replicates and statistic evaluation, in particular when proposing a western-based acetylation signal as a potential biomarker.

We have now provided quantifications for all representative Western blots, where these were missing (Fig. 1b; Fig. 5a, b, Fig. S9). As stated in the journal policy, "Quantitative comparisons between samples on different gels/blots are discouraged" (<http://www.nature.com/authors/policies/image.html>) which is the reason why we have provided independent quantifications for each individual western blot and mentioned in the legend that they are representative of at least $n \geq 3$ independent western blots using at least $n \geq 3$ independent (biological replicate) samples.

In regard to acetylation data, please note that we provide data from patients and mouse cells, as well as from mouse tissues via immunofluorescence and immunohistochemistry. We have also now added additional Western immunoblotting data in Fig. S9a.

- Since the increase in tubulin-acetylation in HGPS patients/models cannot be causally linked to NAT10 via changes in protein levels or enzymatic activity, the value of tubulin-acetylation as a readout for NAT10 function remains questionable. Please discuss this carefully.

We thank the referee for this interesting comment. Indeed, we do not see a consistent increase in NAT10 protein levels in HGPS cells lines or mouse tissues. Instead, the data we have support the hypothesis that NAT10 enzymatic activity is higher in HGPS. Unfortunately, at this point, there is no experimental way of directly testing the enzymatic activity of NAT10 in HGPS cells or tissues. Since Tubulin is a known NAT10 substrate, the increased acetyl-Tubulin in HGPS thus appears to be a readout of NAT10 increased activity. This is in accord with our data presented in Figure 5 and in Fig. S9, where we observe a decrease of Tubulin acetylation upon NAT10 inhibition or genetic depletion in HGPS cells or in mouse tissues.

Reviewer #2 (Remarks to the Author):

Balmus et al. report lifespan extension and physiological benefits in a mouse model of the accelerated aging disease Hutchinson-Guilford progeria syndrome by treatment with a small molecule called remodelin, a previously reported putative inhibitor of the protein acetyltransferase NAT10. Remodelin treatment of an established mouse model of HGPS (G609G) leads to a 25% lifespan extension and improvement of cardiac and adipose phenotype. To validate the involvement of NAT10 in HGPS phenotype, the authors generate a NAT10 knockout mouse. Nat10-nulls are embryonic lethal and Nat10 heterozygotes have reduced body weight, altered lipid composition, and hints of sexual dimorphism. Crosses of Nat10^{+/-} with HGPS(G609G) animals showed rescue of several HGPS phenotypes including back curvature, penile relapse, eye keratitis and some cardiac features. Similar effects were observed when G609G-heterozygous mice were used in the crosses. The authors also perform gene expression analysis on G609G-remodelin and G609G/Nat10 animals and identify a set of genes which they suggest may be as biomarkers. Finally, the authors provide evidence for α -tubulin K40 acetylation as a biomarker in HGPS.

The most interesting finding in this study is the demonstration of rescue of some disease symptoms after reduction of NAT10 levels. The effect of remodelin in the HGPS model will be of interest to the field. However, several conclusions are not convincingly supported by the presented data and are largely speculative, there are experimental shortcomings, and critical controls are missing.

We thank the Reviewer 2 for recognizing the inherent interest of our findings and their importance for the field.

The two major weaknesses of this study are:

The first major weakness of the study is that it is not clear whether the observed effect of remodelin is specific to HGPS or represents a general beneficial effect. The longevity study on WT animals was not carried out long enough to assess this critical question. Ideally, the authors would need to test remodelin in a separate accelerated aging model to answer this important question.

While we thank the reviewer for raising this interesting point; we point out that the long-term treatment of WT animals with Remodelin is technically very challenging and would represent a very complex study on its own. Indeed, it would require orally gavaging up to 100 mice or more (at least 50 vector and 50 Remodelin; expecting some will be lost) for 3 or more years (the C57BL/6 mice that we use have a mean longevity of ~550 days with life expectancy of up to 3 years). Oral gavage of the mice is a delicate procedure that might create damage over time and would be very hard to be approved for a study encompassing such a large number of mice for a prolonged 3-year period. Moreover, it would require an enormous amount of Remodelin to treat ~100 animals for their entire lifespan.

While we are ourselves intrigued by the possibility that our compound might delay normal ageing and/or delay age-related pathologies in other progeroid-syndrome models, we suggest that addressing this is beyond the scope of the present study. The current study is targeted at HGPS with the hope that our findings will prompt further studies to explore the potential therapeutic opportunities for these children, which currently have minimal treatment options.

We believe that, in comparison to other studies on progeria mice, we have an unusual extent of *in vivo* data; and note that in the duration of the studies (over 4 years) described in our manuscript (which we estimated cost nearly £500,000), we have used 1448 mice of different genotypes and treatments encompassing:

- Remodelin and Remodelin Fluor treatment;
- PD and PK studies;
- Engineering of a new mouse knockout model for *Nat10*;
- Genetic crosses between *Lmna*^{G609G} mice and *Nat10* mice in both homozygous and heterozygous backgrounds
- Extensive phenotypic (over 300 parameter) analysis of HGPS (homozygous and heterozygous), in both wild-type and *Nat10*^{+/-} backgrounds.

The second major weakness is that the authors indicate that due to animal welfare regulations, animals had to be sacrificed when they had lost 20% of their body weight. These data are presented in figure 1e and 4a but are presented as “survival” data. Although I appreciate the requirement to follow animal welfare policy, the conclusions are presented are highly misleading. The data should be labelled as “time of 20% weight loss”. More importantly, it is not appropriate to conclude that these data can be used as an indicator of survival. The authors

assume that body weight necessarily tracks with survival, but do not provide any data to support this critical assumption. Furthermore, the authors speculate that the lifespan effect would be even larger if measured directly, but we simply do not know and no data to support this extrapolation is provided. No conclusions, including in the title and abstract, regarding lifespan extension should be made. This is particularly true for the Nat10/HGPS cross which only showed a moderate 17% shift in 20% body weight loss.

We understand the reviewer's point of view. In response to this, we have now updated the manuscript throughout using more conservative language (healthspan instead of lifespan) and have clearly indicated the criteria on the survival graphs. We have also removed the extrapolation statement and modified the manuscript title.

We would nevertheless like to point out that, according to previous literature, there is a clear relationship between weight-loss and survival in this mouse model (see for example Osorio et al. 2011 Science Translational Medicine 3(106) Figure 2 b, c; Ocampo A. et al. 2016 Cell 167(7) Figure 4a,b). In all published literature that we could find, progeria mice were not left to die from natural causes (as it is not acceptable based on current animal welfare) but weight was used as a terminal point. Some papers indicate this endpoint as being either 20% or 30% body weight loss, while some other papers do not give a clear indication. That is why we have annotated the graphs as % survival (and clearly indicated the end-point criteria in the legend) and talked about lifespan. As a matter of fact, in the current research environment, survival analyses on any disease does not represent actual survival but animals that are sacrificed at a "humane end-point"; this is true for cancer, ageing or other studies.

Additional points in order of priority:

The quality of data in figure 5a, b is very poor and the presented conclusions are not convincingly supported by these data:

- All western data should be quantified and normalized to tubulin levels or another cellular protein.

We have now provided quantifications for all the representative WB, where these were missing (Fig. 1b; Fig. 5a, b; Fig. S9).

- Multiple WT and HGPS cell lines should be tested.

We have now added these data in Fig. S9.

- In contrast to the authors' statement there appears to be more NAT10 in HGPS cells than controls.

This is indeed the case in some instances. However, we do not observe this increase in NAT10 levels consistently across tissues and cells, which is the reason why we think the increase in tubulin acetylation is linked to an increase enzymatic activity rather than an increase in protein levels (please also see our responses to Reviewer 1).

- The increase in a-tubulin K40ac is not evident in figure 5a. >

This has now been quantified.

- The NAT10 blot in figure 5b is very poor.

Due to our protein extraction techniques, we have not managed to obtain better NAT10 blots from mouse tissues. We apologize for this and would like to point out that NAT10 levels in LmnaG609G/NAT10 +/- mice are better quantified in Figure 3.

- Why is there more lamin C than lamin A in the WT in figure 5b?

This appears to vary between tissues. We note this in the legend to figure 5b.

- Why is there no progerin in G609G/Nat10 +/- animals (figure 5b, lane4).

When looking at high-quality images, we can see the progerin band in lane 4, even though it is rather weak. Hopefully, the reviewer and/or editor will agree with us when looking at the high-quality images.

Does remodelin have an effect on normal aging or is its effect limited to HGPS?

Based on a study that we performed together with the Shanahan's lab (Cobb et al., Aging Cell 2016), we do observe positive impacts of Remodelin on normal ageing vascular smooth muscle cells (decreased DNA damage, nuclear shape defects and senescence). This suggests that Remodelin might indeed also delay aspects of normal ageing.

While we are ourselves intrigued by the possibility that Remodelin might delay normal ageing, we suggest that addressing this is beyond the scope of the present study. As explained in more detail in our responses to Reviewer 1, such work in a wild-type setting would represent a very complex study covering several years, and one that may in fact be infeasible. The current study is targeted at HGPS with the hope that our findings will prompt further studies to explore the potential therapeutic opportunities for these children, which currently have minimal treatment options.

The significance of the gene expression profiles used to identify potential biomarkers is not clear. These proposed markers are not validated and not directly tested.

We understand the reviewer's comment. However, these data are here to support our model rather than to suggest these genes as direct markers, which would require extensive validation that we suggest is out of the scope of the present manuscript.

Figure 1: The authors show rescue of pH2AX by remodelin treatment in G609G mice. DNA damage is one of many nuclear defects. The authors should test for rescue of some of the other prominent nuclear defects.

We have now added nuclear shape analysis as well as total quantification of phospho-H2AX by Western blotting (Fig 1 a, b).

p. 10. Top of page, the reference should be to figure 4g not 5g.

This has now been corrected.

Reviewer #3 (Remarks to the Author):

Comments to the Authors:

This manuscript by Gabriel Balmus et al. reports in vivo studies of a new Nat10 inhibitor Remodelin in LMNAG609G HGPS mice and in LMNAG609G mice with Nat10 partial depletion. The HGPS mice with either Remodelin treatment or partial Nat10 knockdown showed beneficial effects by extending mice lifespan and improving their cardiovascular pathology and function with increased genomic integration reflected by γ H2AX reduction. Those effects via targeting Nat10 are independent of disease-causing progerin expression. As the first-time mouse study for Remodelin, the authors conducted the toxicity evaluation and provided pharmacokinetics profile in WT mouse. They further studied its derivative and found Remodelin-fluor is more effective and less toxic to mice.

Although this study is of great interest in exploring a novel, potential therapeutic remedy for HGPS patients, additional experiments need to be performed to strengthen the main conclusions. Some parts of this manuscript should be described clearly with detailed information and the whole writing should proofread carefully to avoid any confusion to readers. Outlined below are specific comments to help improve this manuscript.

We thank Reviewer 3 for considering our study to be of great interest in exploring a novel, potential therapeutic remedy for HGPS patients.

Major concerns:

1. Skin fibroblasts derived from LMNAG609G/G609G mouse showed less percentage of γ H2AX positive cells upon Remodelin treatment compared to vehicle-treated cells in Figure 1a. The image quality is low and needs to be improved. There is no description for the scale bar in figure legend. Detailed information for how long the cells were treated at what concentration of Remodelin should clearly indicate in figure legend or Method (same for Figure 5a-5b).

We have now added this information in the respective figure legends.

The quantification of γ H2AX-positively stained cells collected by manual counting in histogram is likely subjective based on how to define "Positive cells". Western blotting analysis for γ H2AX expression level normalized to total H2AX should be more accurate to provide quantitative results as shown in Figure 2e.

We thank the reviewer for this suggestion. We have now added a new Western immunoblot as Figure 1b. All the treatment data are referred to in the Methods section but have also now been added into the Figure legends. We have also returned to the original (non-inverted) immunofluorescent images that will give the reviewer a better idea on what a positive cell is. We are sorry that the original images created confusion.

2. Which administration PO or IV was performed for calculating pharmacokinetic (PK) parameters presented in Figure 1c? The description in corresponding figure legend is quite confusing. Please clarify.

PO was used to calculate PK as it showed a better profile. We have now added this information on top of the panel Figure 1d.

How long after PO administration of high dose Remodelin were the tissues collected for mass spectrometry assay in Figure 1d?

Tissues were collected after 2 weeks' of daily dosing, and 1 hour after the last dosing for mass spectrometry analysis. We have now added this information in the figure legend.

3. PK studies usually show gender-based differences. Are there any specific reasons to use mice with different gender in two sets of PK studies in Table S1 and S2?

We understand the reviewer's point. The scientific group from Crown Biosciences (that we have accessed because of their extensive experience with PK studies) has fed back to us that preclinical studies intended to be included in files for regulatory submission may be run on both gender in the same study. Although strictly speaking, PK

studies should use animals of both sexes (gender-based differences in metabolism are believed to be the main cause of differential pharmacokinetics between male and female), the recent advancement in the regulation of use of animals in research (3R's – Replacement; Reduction; Refinement) shifted the approach to the use of the current experimental design where males are used in one round and females in another round. This experimental design allows for better statistical power (thus overall allows for the reduction of the number of animals used) and will expose any sex bias (if inconsistencies are detected between the two cohorts there would be reason to expand by performing additional experiments to understand any sexual dimorphism). Based on the data presented, there was no reason to believe there was sex-based differences.

There is no PK result in the entire manuscript that is corresponding to Table S2 and it is very important to check Remodelin concentrations in heart and muscle after PO or IV dosing at 100 mg/kg. Have the authors conducted PK study in LMNAG609G/G609G HGPS mice and checked Remodelin concentrations in their main organs, including heart/aorta, liver, kidney, lung, brain, bone, and skin? The accumulation of Remodelin after multiple or long-term administrations in different tissue and organ would probably provide some explanations for their pathological phenotype changes.

We thank the reviewer for raising this point. We have now added new data obtained from HGPS mice in Fig. S1c; please note the detailed legend for this figure indicating the amount of Remodelin present in the plasma in WT and HGPS mice upon 1 hour and 24 hours' treatment. These data show that Remodelin does accumulate in the tissues of the HGPS mice upon PO dosing of 100 mg/kg Remodelin both in muscle and liver but also in the brain (indicating that Remodelin can pass the blood/brain barrier).

4. The description for Remodelin toxicity assessment is confusing. If I understand correctly, the authors conducted a sub-acute toxicity trial (Figure S1a) after repeated Remodelin administration by PO or SC for two-week duration, and a chronic toxicity trial by PO dosing Remodelin at 100 mg/kg/day for 13 weeks (Figure S1b) and Remodlin-fluor at 50 mg/kg/day. They claimed that this new chemical at 100 mg/kg/day is well tolerated without body weight loss except "the only observed side effect being hair greying". Since it a brand new chemical, further study for its toxicity is highly suggested, e.g. acute toxicity assessment to determine LD50 (Median Lethal Dose) and MTD (Maximum Tolerated Dose) that will help optimize the ideal dosage for lifespan trial, the histopathology examination in main organs like liver, kidney, lung, etc. in chronic toxicity trial, etc.

We are sorry that we have not been clear enough about the toxicity assessment. As the reviewer notes, Remodelin and derivatives are indeed new chemicals that had not been tested *in vivo* in previous studies.

For this reason, we indeed performed an initial sub-acute toxicity assessment to determine the range of concentrations at which the molecule was well tolerated. Based on this study, on the PK results and the mass spectrometry quantification of the molecule in tissues presented in Figure 1c-e, we decided to treat the progeria mice by daily oral gavage at 100mg/kg/day. Since we obtained good phenotypes improvement at this dose without side effects, we decided to carry out the study with this dosage. We are not claiming that we have optimized the drug regimen for human pre-clinical studies or lifespan trials, as this is something that will have to be tested by a company in very specific assays before the molecule could go forward. We do not have the assays in place to carry out this kind of studies in the lab and we feel that this is out of the scope of the current study.

5. Authors showed pathological improvement in G609G mouse skin and aorta after Remodelin treatment in Figure 2a at endpoint of individual mouse. It's confusing. Please clarify the specific endpoint for each mouse in Figure 2a-2d. Normally, such comparison should be conducted at same time point after drug treatment.

We have now added a new table (Table S4) that identifies the age of the mice used in these comparisons. It can be noted that the Remodelin treated mice are older than the vehicle treated mice. We have used the end-point to be consistent with previously published studies, but more importantly to be conservative in our analyses. Because Remodelin treated mice were showing improved health at the 20% body weight loss end-point (please see supplementary movies S1, S2, S3), we wanted to support these data with the pathological data at endpoint.

6. Figure 2e: Please explain why there was no visible lamin A band in heart tissues from G609G mouse with/without Remodelin treatment, and why the lamin C in two vehicle-treated samples had a clear band shifting compared to WT and treated sample. If this is due to some technical issues, please provide better representative WB data.

This is a good observation but please note that, in contrast to patients that are heterozygous for the *Lmna* mutation, the mice here have a homozygous G609G mutation. Therefore, they do not express mature Lamin A, but only

progerin. Regarding the band shifting, we do indeed observe this in some tissues, but we do not have an explanation for it at this stage.

7. Authors claim the molecular mechanism underlying Remodelin effects partly through deacetylation of α -tubulin (Figure 5 and Page 14 Paragraph 2): “Remodelin treatment decreases the high α -tubulin K40 acetylation in HGPS-patient derived cells (a) and mouse tissues (b).” and “... indicating that NAT10 inhibition ameliorates HGPS cellular phenotypes, at least in part by mediating microtubule destabilization.” How to explain the elevated level of acetylated α -tubulin at lys-40 in HGPS cells and mouse tissues? Nat10 expression actually looks higher in HGPS fibroblasts compared to Healthy control (Figure 5a), quantitative analysis should be provided to compare its expression between disease and control cells. What’s Nat10 activity in disease cells and tissues vs. control? What’s the functional correlation of α -tubulin K40 acetylation to nuclear morphology?

We thank the referee for this interesting comment. As mentioned in our responses to Referee 1, we indeed do not see a consistent increase in NAT10 protein levels in HGPS cells lines or mouse tissues. Instead, the data we have support the hypothesis that NAT10 enzymatic activity is higher in HGPS. Unfortunately, at this point, there is no experimental way of directly testing the enzymatic activity of NAT10 in HGPS cells or tissues. Since Tubulin is a known NAT10 substrate, the increased acetyl-Tubulin in HGPS thus appears to be a readout of NAT10 increased activity. This is in accord with our data presented in Figure 5 and in Fig. S9, where we observe a decrease of Tubulin acetylation upon NAT10 inhibition or genetic depletion in HGPS cells or in mouse tissues.

8. Authors state (Page 12): “... Remodelin leads to lifespan and fitness improvements in both homozygous and heterozygous HGPS mice via a mechanism that appears to be independent of progerin.” Previous study from same group also indicates Remodelin-induced improvement of the nuclear shape in HGPS cells without reduction of progerin expression. Are there any changes for progerin distribution in HGPS nucleus after Remodelin treatment? The redistribution of progerin from nuclear rim into nucleoplasm could improve nuclear morphology due to less tethering of progerin to nuclear envelope. Immunofluorescence staining of progerin protein in HGPS cells upon Remodelin treatment would seem adequate to exclude this situation.

We thank the reviewer for this interesting comment. However, in our experiments, we have never observed changes in progerin localization upon Remodelin treatment or NAT10 depletion. Below, we provide examples of representative images from one of the HGPS cell lines (please note that we observe similar progerin localization in other HGPS cell lines). Therefore, we believe that NAT10 inhibition does not work via modifying progerin expression or localization.

Minor Concerns:

There are many wrong references in the entire manuscript including supplementary materials. Please proofread carefully and rearrange or correct them.

We thank the reviewer for the comments. We have now updated all the figures/text that were incorrectly-referenced.

For example:

1) Supplementary Material Page 7: The title is “Pharmacokinetics Evaluation ...”. It is actually more about lifespan study design (Table S3, Paragraph 198~199) but not for PK study (Table S1 & S2). The description for MS assay (Page 6 last paragraph) was put in toxicity session.

2) Supplementary Material Page 23 Figure S2: “Identification of a more potent Remodelin analogue”. However, this S2 is about the coronary pathology.

REVIEWERS' COMMENTS:

Reviewer #2 (Remarks to the Author):

The revision by Balmus et al. partially address our original points. Several however remain:

The authors argue that they can not perform long term studies due to experimental complexities and large amounts of compound required. I appreciate that point, but the inability to perform these experiments does not justify conclusions regarding normal aging (see discussion) that are not experimentally tested. I also did not understand the argument that large amounts of remodelin can not be generated, which would seem to preclude any clinical studies the authors repeatedly refer to.

I maintain our point that the "survival" studies are not survival studies but loss of body weight studies. It is the latter parameter that is measured and any conclusions regarding longevity is inference. The change to "healthspan" is equally inappropriate as this term refers to a combination of parameters, but not a single parameter as measured here. I suggest the authors report accurately on what they actually measured.

The section on "biomarkers" is still very weak and relies entirely on correlation. The "biomarkers" are not validated in any way. The same applies to the tubulin acetylation section. The authors argue in their rebuttal for several points that there is variability in NAT10 and lamin isoforms between tissues and animals, yet this variability which precludes them from providing these datasets, is not addressed in their quest for biomarkers. If the authors want to claim they found biomarkers for HGPS they should do the proper validation experiments; in the absence of such data these claims should be removed.

Reviewer #3 (Remarks to the Author):

The authors are to be commended for their efforts to address the issues raised in my previous report. I only have minor comments to improve this excellent work.

1. The unit of Remodelin concentrations in mouse tissues is "ng/mL" in Figure 1e while it is "ng/g" in Figure S1C. Please make sure whether this is a typo or additional conversion between different units is needed.

2. Please use clear fonts in Figure 5b legend & 5d image labeling.

Reviewer #2 (Remarks to the Author):

The revision by Balmus et al. partially address our original points.

Several however remain:

The authors argue that they can not perform long term studies due to experimental complexities and large amounts of compound required. I appreciate that point, but the inability to perform these experiments does not justify conclusions regarding normal aging (see discussion) that are not experimentally tested.

We thank the reviewer for understanding the technical issues of long term studies. We have now toned down our conclusions regarding normal aging, and clearly state that this is just a speculation.

I also did not understand the argument that large amounts of remodelin can not be generated, which would seem to preclude any clinical studies the authors repeatedly refer to.

It would actually be very feasible to synthesize large amounts of Remodelin for clinical studies. It is the technical difficulties to orally gavage the mice that preclude such in vivo studies. We are sorry if we generated confusion in our previous discussion with the reviewer.

I maintain our point that the “survival” studies are not survival studies but loss of body weight studies. It is the latter parameter that is measured and any conclusions regarding longevity is inference. The change to “healthspan” is equally inappropriate as this term refers to a combination of parameters, but not a single parameter as measured here. I suggest the authors report accurately on what they actually measured.

We have now toned down our claims about survival and healthspan in the text and removed any direct reference to survival. We instead refer to “increased time frame of body weight loss”.

The section on “biomarkers” is still very weak and relies entirely on correlation. The “biomarkers” are not validated in any way. The same applies to the tubulin acetylation section. The authors argue in their rebuttal for several points that there is variability in NAT10 and lamin isoforms between tissues and animals, yet this variability which precludes them from providing these datasets, is not addressed in their quest for biomarkers. If the authors want to claim they found biomarkers for HGPS they should do the proper validation experiments; in the absence of such data these claims should be removed.

We have now removed the word “biomarker” and toned down our claims regarding the values of these potential biomarkers. We now refer to these as “potential readouts” for NAT10 inhibition in vivo and clearly state that further investigation would be required to assess their potential as biomarkers.

Reviewer #3 (Remarks to the Author):

The authors are to be commended for their efforts to address the issues raised in my previous report. I only have minor comments to improve this excellent work.

We thank the reviewer for acknowledging the changes we have made in our revised manuscript.

1. The unit of Remodelin concentrations in mouse tissues is “ng/mL” in Figure 1e while it is “ng/g” in Figure S1C. Please make sure whether this is a typo or additional conversion between different units is needed.

Thank you for pointing that out. The correct unit in Figure 1e is nM and not ng per mL – we have now changed that. In Supplementary Figure 1C, the unit of ng per g of tissue is correct.

2. Please use clear fonts in Figure 5b legend & 5d image labeling.

We have now modified the fonts in these figures.